# Intranasal Administration of Mesenchymal Stem Cell Secretome Reduces Hippocampal Oxidative Stress, Neuroinflammation and Cell Death, Improving the Behavioral Outcome Following Perinatal Asphyxia

**DOI:** 10.3390/ijms21207800

**Published:** 2020-10-21

**Authors:** Nancy Farfán, Jaime Carril, Martina Redel, Marta Zamorano, Maureen Araya, Estephania Monzón, Raúl Alvarado, Norton Contreras, Andrea Tapia-Bustos, María Elena Quintanilla, Fernando Ezquer, José Luis Valdés, Yedy Israel, Mario Herrera-Marschitz, Paola Morales

**Affiliations:** 1Molecular & Clinical Pharmacology Program, ICBM, Faculty of Medicine University of Chile, Santiago 8380453, Chile; nancy.farfan.t@gmail.com (N.F.); carril.jaime@gmail.com (J.C.); martina.redel90@gmail.com (M.R.); zamorano.c.marta@gmail.com (M.Z.); arayapacheco.m@gmail.com (M.A.); estephaniamon.zon@gmail.com (E.M.); r.alvaradorosas@gmail.com (R.A.); equintanilla@med.uchile.cl (M.E.Q.); yisrael@uchile.cl (Y.I.); mh-marschitz@med.uchile.cl (M.H.-M.); 2Department of Neuroscience, Faculty of Medicine, University of Chile, Santiago 8380453, Chile; ncparedes@gmail.com (N.C.); jlvaldes@uchile.cl (J.L.V.); 3School of Pharmacy, Faculty of Medicine, Universidad Andres Bello, Santiago 8370149, Chile; ac.tapiabustos@gmail.com; 4Center for Regenerative Medicine, Faculty of Medicine-Clínica Alemana, Universidad del Desarrollo, Santiago 7710162, Chile; eezquer@udd.cl

**Keywords:** Neonatal hypoxia, mesenchymal stem cell secretome (MSC-S), intranasal administration, oxidative stress, neuroinflammation, cell death, behavioral development, hippocampus, memory, neuroprotection

## Abstract

Perinatal Asphyxia (PA) is a leading cause of motor and neuropsychiatric disability associated with sustained oxidative stress, neuroinflammation, and cell death, affecting brain development. Based on a rat model of global PA, we investigated the neuroprotective effect of intranasally administered secretome, derived from human adipose mesenchymal stem cells (MSC-S), preconditioned with either deferoxamine (an hypoxia-mimetic) or TNF-α+IFN-γ (pro-inflammatory cytokines). PA was generated by immersing fetus-containing uterine horns in a water bath at 37 °C for 21 min. Thereafter, 16 μL of MSC-S (containing 6 μg of protein derived from 2 × 10^5^ preconditioned-MSC), or vehicle, were intranasally administered 2 h after birth to asphyxia-exposed and control rats, evaluated at postnatal day (P) 7. Alternatively, pups received a dose of either preconditioned MSC-S or vehicle, both at 2 h and P7, and were evaluated at P14, P30, and P60. The preconditioned MSC-S treatment (i) reversed asphyxia-induced oxidative stress in the hippocampus (oxidized/reduced glutathione); (ii) increased antioxidative Nuclear Erythroid 2-Related Factor 2 (NRF2) translocation; (iii) increased NQO1 antioxidant protein; (iv) reduced neuroinflammation (decreasing nuclearNF-κB/p65 levels and microglial reactivity); (v) decreased cleaved-caspase-3 cell-death; (vi) improved righting reflex, negative geotaxis, cliff aversion, locomotor activity, anxiety, motor coordination, and recognition memory. Overall, the study demonstrates that intranasal administration of preconditioned MSC-S is a novel therapeutic strategy that prevents the long-term effects of perinatal asphyxia.

## 1. Introduction 

Perinatal Asphyxia (PA), and resulting hypoxic-ischemic encephalopathy (HIE), is a major obstetric complication occurring during labor, constituting a leading cause of neonatal death [1,2]. Surviving children develop long-lasting motor and learning disabilities, as well as neuropsychiatric and cognitive deficits such as cerebral palsy, mental retardation, epilepsy, hyperactivity, autism, attention deficit, schizophrenia, and psychiatric disorders [1,2]. At present, there is no effective treatment to prevent these sequelae [3]. Worldwide, each year, over one million children develop acute neuro-disabilities [4] leading to major psychosocial and socioeconomic consequences for the individual and the national budgets. Thus, protecting the newborn brain from injury at birth is a global health priority. 

PA implies an impaired gas exchange, progressive hypoxemia, hypercapnia, and acidosis, occurring in utero, during labor, and/or delivery [2,4]. PA generates a primary energy crisis because of the oxygen deficit, followed by a secondary insult due to re-oxygenation, and long-lasting oxidative stress, neuroinflammation and cell death, affecting specific brain regions, including the hippocampus [5,6,7], basal ganglia [8], and white matter [9]. 

The energy deficit alters brain metabolism, triggering cascades of events generating an imbalance of free radicals, reactive nitrogen (RNS) and oxygen (ROS) species, and pro-inflammatory molecules [10,11,12]. These cascades affect brain areas differently, depending on the energy requirements and the developmental stages of different regions at the time of the insult, defining settings of vulnerability [13], but also allowing a window of therapeutic opportunity. 

The neonatal brain is highly vulnerable to oxidative stress, due to its low levels of antioxidant defenses, including low expression/activity of the superoxide dismutase (SOD) family, reduced glutathione (GSH), glutathione peroxidase (GPx), catalase and peroxiredoxin-3 (Prx-3) [14,15]. Furthermore, at birth, the brain is characterized by high oxygen consumption, high unsaturated fatty acid content [15,16], high water content, and high availability of redox-active iron [17,18]. 

We have recently reported that PA induces sustained oxidative stress, reflected by a long-lasting increase of the oxidized glutathione (GSSG) level, and a high GSSG/GSH ratio, together with a decrease in tissue reducing capacity and catalase activity, resulting in apoptotic caspase-3-dependent cell death, mainly in the basal ganglia and hippocampus [7,19]. Endogenous repair mechanisms such as neurogenesis, synaptogenesis, and anti-apoptotic pathways have been suggested to be activated, but these are not always sufficient to resolve the deficit [5,13,20,21]. 

Nuclear Erythroid 2-Related Factor 2 (NRF2) is an important protective transcription factor, known as a master regulator of antioxidant defenses [22]. Under homeostatic conditions, NRF2 is sequestered in the cytoplasm by its inhibitor Kelch-like ECH-associated protein 1 (Keap1), an adapter for the scaffolding protein Cullin 3 (Cul3), which binds the E3 ligase complex to ubiquitination and proteasomal degradation, maintaining low cytosolic NRF2 levels. Following oxidative stress, NRF2 is activated and translocated into the nucleus, where it binds to the antioxidant/electrophilic response element (ARE), regulating the expression of multiple cytoprotective antioxidant and glutathione generating enzymes, including NAD(P)H dehydrogenase (quinone 1) (NQO1), heme oxygenase-1 (HO1), SODs, catalase, glutathione S-transferase (GST), GSH reductase [22,23], reducing ROS levels, and maintaining the homeostasis [24]. The role of NRF2 in the long-term oxidative stress elicited by PA requires further investigation.

Neuroinflammation generated by oxidative stress is another key player of PA pathology. ROS lead to nuclear translocation of the transcription factor nuclear factor-kappa B (NF-κB) subunit p65, overexpressing the pro-inflammatory cytokines IL-1β and TNF-α, leading to glial reactivity and cell death [25]. Microglia and astrocytes are the cells firstly activated by PA, as evidenced by morphological changes, migration to damaged regions, and release of: (i) pro-inflammatory cytokines, TNF-α, IL1-β, and IL-6; (ii) glutamate; (iii) nitric oxide, and (iv) additional free radicals [26,27,28], contributing to a harmful cellular environment, causing neuronal and glial death [29,30]. Microglial infiltration, astrogliosis and upregulation of TNF-α and IL-1β have been observed 24 h and up to 2 months after birth in white and grey matter of post-mortem brains of infants suffering from hypoxic-ischemic encephalopathy [31,32]. Moreover, NF-κB /p65 translocation, IL-1β and TNF-α overexpression, and apoptotic-like cell death were found 8 and 24 h after PA in the mesencephalon and hippocampus of rat neonates [10,25]. 

Pro-inflammatory cytokines released by reactive glial cells also produce cytotoxic effects on oligodendrocytes [33], inducing premature maturation of oligodendrocyte progenitor cells and aberrant myelination, thereby contributing to hypomyelination in the developing brain [34], and to delays in neurobehavioral development [35,36], and cognitive impairments [2,37], including deficit in recognition memory [35,38], social skills [39], and increased anxiety [6,35], as well as deficits in motor coordination [6,35]. 

Overall, cell damage following PA is associated and potentiated by oxidative stress and inflammatory signaling, constituting major therapeutic targets [4,7,10,40].

Cognitive and motor dysfunction have also been observed in the PA model used in this study. It was shown that PA impaired nonspatial memory (assessed by novel object recognition) and motor coordination (assessed by the rotarod test) in adult rats, demonstrating PA-induced long-term effects on neurobehavioral functions [35,36]. PA pups showed delayed emergence of several reflexes, including righting, negative geotaxis, fore- and hind-limb grasping and placing [36]. These deficits require therapeutic interventions to improve neurobehavioral development [41].

Mesenchymal stem cells (MSC) have been proposed as a therapeutic tool for neonatal central nervous system (CNS) diseases [42,43] due to their broad range of anti-inflammatory, anti-oxidant, and pro-regenerative effects, involving both paracrine and cell-to-cell contact mechanisms [44,45,46]. 

The long-lasting anti-inflammatory and antioxidant effects of MSC are well documented in adult animal models of stroke, ischemia, traumatic brain injury [47,48], and chronic alcohol intake [49]. Recent reports from our group showed that a single intra-cerebro-ventricular (i.c.v.) administration of MSC to rats after PA prevented the loss of mature oligodendrocytes and delayed myelination in telencephalic white matter [50].

It is noteworthy that live MSC administration entails a risk for translational studies. Currently, the use of secretomes derived from MSC (MSC-S) has been promoted, because the secretome contains a set of bioactive factors secreted into the extracellular space. MSC-S can be evaluated for safety, dosage, and potency similarly to conventional pharmaceutical drugs, and is readily available since it can be lyophilized, favoring storage, and implying low factoring costs for large-scale production reutilizing the yielding cells [51].

Preconditioning approaches to enhance MSC therapeutic efficacy, thereby generating cellular products with improved potential for different clinical applications, have been developed. In vitro preconditioning of MSC with cytokines and growth factors, hypoxia, drugs, biomaterials, and different culture conditions, including the iron chelator deferoxamine (DFX), have led to improved MSC-S therapeutic strategies [52]. DFX is a hypoxia-mimetic agent used to precondition adipose tissue-derived MSC (AD-MSC), increasing the production and secretion of pro-angiogenic, neuroprotective, antioxidant, and anti-inflammatory molecules [53,54]. It has been demonstrated that systemic administration of conditioned medium derived from DFX-preconditioned MSC reduced sensory neuron and Schwann cell apoptosis, improving angiogenesis, reducing chronic inflammation, and increasing neuroprotective effects in a diabetic polyneuropathy mouse model [54], suggesting that this therapeutic intervention may be effective to prevent PA’s deleterious effects.

Recent studies have shown that AD-MSC, preconditioned with pro-inflammatory cytokines, improved anti-inflammatory, pro-regenerative, and anti-oxidative properties [53,55]. The combination of pro-inflammatory cytokines, such as TNF-α+IFN-γ used for MSC preconditioning, led to additional beneficial effects, inhibiting complement activation by increasing factor H production, resulting in a decreased immune response [52,56]. Thus, pro-inflammatory MSC preconditioning is a strategy to improve immunosuppressive responses, increasing secretion of anti-inflammatory and immunomodulatory factors [52,57], emerging as a potent therapeutic tool against inflammatory disease. Indeed, in a translational study, the noninvasive intranasal administration route of the secretome generated from human AD-MSC preconditioned with TNF-α+IFN-γ (MSC-S) fully reversed alcohol-induced hippocampal oxidative stress and neuroinflammation [58].

No studies have been conducted to date to test the effects of intranasal MSC-S administration in PA. Hence, this study investigates the efficacy of intranasal administration of preconditioned MSC-S in reducing oxidative stress, neuroinflammation, and cell death in the hippocampus, and in mitigating the behavioral, motor, and cognitive disabilities elicited by PA. Aiming at the mechanism of MSC-S action, the study further explores the role of the NFR2 activation pathway in reducing ROS levels and improving the homeostasis vulnerated by the perinatal metabolic insult.

## 2. Results

### 2.1. Apgar Scale

Table 1 shows the outcome of PA evaluated by an Apgar scale adapted for rats, applied 40 min after delivery (the time of the uterine excision). Control pups initiated pulmonary breathing as soon as the amniotic tissue was removed from the animal’s nose, whereas asphyxia-exposed rats had to be repetitively stimulated to respire until a first breath was supported by forced gasping up to stabilization. The rate of survival shown by PA rat neonates was approximately 66%, while it was 100% among controls. Surviving asphyxia-exposed rats showed decreased respiratory frequency (~60%), decreased vocalization (~70%), blue (cyanotic) skin, rigidity, and akinesia, compared to sibling control rats, indicating a severe insult. Thereafter, controls and asphyxia-exposed rats were randomly assigned to each experimental condition. After intranasal administration of vehicle or preconditioned MSC-S all the animals survived.

### 2.2. Effect of Intranasal Administration of Secretome Derived from Preconditioned Human Adipose MSC (MSC-S) on Hippocampal Oxidative Stress Induced by PA, Evaluated at P7 and P60

16 μL of vehicle (saline) or MSC-S (containing 6 μg of protein from 2 × 10^5^ MSC), preconditioned with either DFX (DFX-MSC-S) or TNF-α+IFN-γ (TNF-α+IFN-γ-MSC-S), were intranasally administered two hours after birth to asphyxia-exposed and control rats, and the effect on the oxidized/reduced glutathione ratio (GSSG/GSH; an indicator of oxidative stress) was evaluated at P7. To study the long-term protection against hippocampal oxidative stress, two doses of MSC-S were administered, the first two hours after birth, and the second at P7, evaluating the effects up to P60 (Figure 1A).

As shown in Figure 1B, PA increased the GSSG/GSH ratio > 3.5-fold (*p* < 0.0005, *N* = 9) at P7, and > 2-fold (*p* < 0.0005, *N* = 4) at P60 (Figure 1C), as compared to those observed in vehicle treated control animals (CS) (N = 9; *N* = 5, respectively).

Intranasal administration of DFX-MSC-S fully reversed the effect of PA on the GSSG/GSH ratio observed at P7 (*p* < 0.005, *N* = 10; Figure 1B) and at P60 (*p* < 0.0001, *N* = 5; Figure 1C). A similar effect was produced by TNF-α + IFN-γ-MSC-S, observed at P7 (*p* < 0.005, *N* = 6, Figure 1B) and at P60 (*p* < 0.0001, *N* = 6; Figure 1C), as compared to that observed in vehicle-treated asphyxia-exposed animals (AS).

### 2.3. Effect of Intranasal Administration of MSC-S on Hippocampal Cytoplasmic and Nuclear NRF2 Protein Levels in PA-Exposed and Control Rats at P7

AS rats showed increased cytoplasmic NRF2 protein levels (>2-fold; *p* < 0.05, *N* = 7), compared to those by CS (*N* = 7; Figure 2A,C). A single intranasal administration of DFX-MSC-S (*p* < 0.005, *N* = 6) or TNF-α+IFN-γ-MSC-S (*p* < 0.05, *N* = 7) reversed the effect of PA on cytoplasmic NRF2 protein levels (Figure 2A,C).

Increased NRF2 nuclear translocation was observed in the hippocampus of PA rats treated either with DFX-MSC-S (>4-fold; *p* < 0.005, *N* = 7) or TNF-α+IFN-γ-MSC-S (>4-fold; *p* < 0.05, *N* = 8), compared to AS rats (Figure 2B,D), also when compared to CS rats (>3-fold; *p* < 0.05, *N* = 9; Figure 2B,D). The effect was associated with a decrease of cytoplasmic NRF2 levels in PA-exposed rats, treated with either DFX-MSC-S, or TNF-α+IFN-γ-MSC-S (Figure 2A,C). No differences in nuclear and cytoplasmic NRF2 protein levels were found when comparing MSC-S-treated versus vehicle-treated controls (Appendix A).

To explore the downstream cascade of the NRF2 pathway, hippocampal cytoplasmic NAD(P)H quinone oxidoreductase (NQO1) and heme oxygenase-1 (HO1) protein levels were analyzed by Western blot. As shown by Figure 3A,C, no differences were observed in NQO1 levels, between AS versus CS rats at P7. Nevertheless, a single intranasal DFX-MSC-S administration to PA-exposed rats increased NQO1 levels, compared to both AS (>2.5-fold; *p* < 0.005, *N* = 6) and CS (>2-fold; *p* < 0.05, *N* = 6) rat neonates. In contrast, TNF-α+IFN-γ-MSC-S administration did not produce any effect on NQO1 levels, compared to AS or CS rats. The effect of DFX-MSC-S was significant when compared to that observed after TNF-α+IFN-γ-MSC-S administration in PA-animals (>1.5-fold; *p* < 0.05, *N* = 6) (Figure 3A,C).

No effect of PA and/or MSC-S treatment was observed on HO1 protein levels, evaluated in hippocampus at P7 (*N* = 8–9, for each condition; Figure 3B,D). No differences were observed on NQO1 and/or HO1 levels in control rats treated either with DFX-MSC-S or TNF-α+IFN-γ-MSC-S, compared to CS (Appendix A).

### 2.4. Effect of Intranasal Administration of MSC-S on PA-Induced Neuroinflammation Determined in Hippocampus at P7

The effect of PA and DFX- or TNF-α+IFN-γ-MSC-S administration on NF-κB/p65 nuclear translocation was evaluated in the hippocampus from control and PA-exposed rats at P7. A reduction in nuclear p65 was expected to reduce neuroinflammation, *vide infra.*
Figure 4 shows that nuclear p65 protein levels were decreased when PA-exposed rats were treated with either DFX-MSC-S (by 80%; *p* < 0.0001, *N* = 6) or TNF-α+IFN-γ-MSC-S (by 82%; *p* < 0.0001, *N* = 4), compared to AS (*N* = 6) or CS (*N* = 6) rats (Figure 4B,D). However, no differences were observed on cytoplasmic p65 levels among any of the experimental groups, except for an increase of p65 levels produced by TNF-α+IFN-γ-MSC-S-treatment in PA-exposed neonates (>2-fold; *p* < 0.05), compared to CS (Figure 4A,C, *N* = 6 each experimental condition). No differences in nuclear and cytoplasmic p65 levels were found among DFX-MSC-S- or TNF-α+IFN-γ-MSC-S treated control rats, compared to CS rats (Appendix A).

### 2.5. Effect of Intranasal Administration of MSC-S on PA-Induced Microglial Reactivity Determined in Hippocampus at P7

Microglial reactivity was observed as a retraction of microglial processes. The length of primary and secondary microglial processes in the *stratum radiatum* of the CA1 hippocampal region at P7 was analyzed using the microglial marker calcium binding adaptor molecule 1 (IBA-1) together with the nuclear marker DAPI. As seen in Figure 5, PA caused a decrease in the length of primary (by 53 %, *p* < 0.005; *N* = 14), and secondary (by 48%, *p* < 0.005; *N* = 21) microglial processes (IBA-1 immunofluorescence; red) in the hippocampus of AS compared to that observed in CS neonatal rats (*N* = 14), suggesting microglial reactivity, a neuroinflammatory indication induced by PA. Intranasal administration of MSC-S reversed this effect, both on primary (*p* < 0.005; *N* = 18) and secondary microglial process length (*p* < 0.05; *N* = 14), compared to AS rats (Figure 5B,C).

### 2.6. Effect of Intranasal Administration of MSC-S on Hippocampal Apoptotic-Like Cell Death (Cleaved-Caspase-3) Induced by PA at P7

The effect of PA and DFX- or TNF-α+IFN-γ-MSC-S administration on apoptotic cell death was evaluated by assessing cleaved-caspase-3 protein levels in hippocampus from control and PA-exposed rats at P7 (Figure 6). PA induced an increase of cleaved-caspase-3 levels (> 3-fold; *p* < 0.005, *N* = 6) (AS), compared to that shown by CS rats (*N* = 6). Intranasal administration of DFX-MSC-S (*N* = 6; *p* < 0.05) or TNF-α+IFN-γ-MSC-S (*N* = 6; *p* < 0.05) fully reversed the effect of PA on cleaved-caspase-3.

### 2.7. Effect of Intranasal Administration of MSC-S on the Behavioral Outcome Induced by PA

The effect of PA and DFX- or TNF-α+IFN-γ-MSC-S-administration (single dose at 2 h after birth or two doses at 2 h and P7) was evaluated on behavioral development, monitored by the righting, cliff aversion, and negative geotaxis reflexes, motor coordination, locomotor activity, recognition memory, and anxiety (see Appendix A). The tests were performed considering the occurrence of the respective reflexes and behavioral action programs along postnatal development, in accordance with previous reports [59,60].

The righting reflex was evaluated by monitoring the time required to recover from a forced supine to a natural prone position, as indicative of brain stem and spinal cord reflexes [61]. Righting evaluation at P1 showed that AS rats remained for a longer time (sec) in the supine position (~1.5-fold; *p* < 0.005, *N* = 36) than CS rats (*N* = 36; Figure 7A). TNF-α+IFN-γ- MSC-S administration improved the righting reflex (*p* < 0.05; *N* = 38) to values similar to those shown by CS animals (Figure 7A). No effect on the righting reflex was observed in DFX-MSC-S-treated PA-exposed rats.

Cliff aversion and negative geotaxis were evaluated at P14, for monitoring sensorimotor functions [61]. To evaluate cliff aversion, rats were placed on an elevated flat ledge with their forelimbs and head positioned over the edge. The time it took for the rats to move away from the cliff and to move their legs and head away from the edge was recorded. It was found that AS rats increased their lag time (sec) in the cliff aversion test (>4-fold; *p* < 0.05, *N* = 8), compared to that shown by CS rats (Figure 7B, *N* = 9). The effect of PA on cliff aversion was reversed by DFX-MSC-S (*p* < 0.005, *N* = 7) or TNF-α+IFN-γ-MSC-S (*p* < 0.05, *N* = 9) (Figure 7B).

To test for negative geotaxis, the rats were placed face down on a 30° inclined grid. The hindlimbs of the rat were placed in the middle of the grid. The time (sec) at which the rats turned to face up the slope and to climb up the grid with their forelimbs reaching the upper rim was recorded. It was observed that AS rats increased their time in negative geotaxis (~2-fold; *p* < 0.05; *N* = 8), compared to that shown by CS rats (Figure 7C; *N* = 8). Only DFX-MSC-S (*N* = 5) reversed the effect of PA (*p* < 0.005), compared to that shown by AS rats (Figure 7C).

Motor coordination was evaluated as the time (sec) spent on a rotarod device at P60. PA-exposed rats showed shorter retention time (by 30%, *p* < 0.0001), compared to that shown by CS rats (Figure 7D). The effect of PA was reversed by both DFX-MSC-S (*p* < 0.005) and TNF-α+IFN-γ-MSC-S (*p* < 0.005) treatment.

Exploratory behavior was estimated by measuring locomotor activity (cm) in a square arena (open field) at P7 and P30. As shown by Figure 8A, locomotor activity was decreased in AS (by 40%; *p* < 0.05, *N* = 10), compared to that in CS rats at P7. Only DFX-MSC-S administration improved locomotor activity shown by PA-exposed rats (*p* < 0.05, *N* = 10), to levels similar to that shown by CS rats (*N* = 9; Figure 8A). AS and PA TNF-α+IFN-γ-MSC-S-treated (*N* = 8) rats showed less locomotion, compared to CS and PA-DFX-MSC-S treated groups (Figure 8A). Similar results were observed at P30, when locomotor activity shown by PA-exposed (AS) rats was still decreased (by 45%, *p* < 0.005, *N* = 12), compared to CS animals (*N* = 12, Figure 8C). DXF-MSC-S administration reversed the effect of PA (*p* < 0.05, *N* = 10), while TNF-α+IFN-γ-MSC-S did not reverse the effect of PA on locomotor activity (*N* = 8; Figure 8C). Representative tracking by vehicle- and MSC-S-treated rats and their occupancy maps in the arena are shown at P7 (Figure 8B) and P30 (Figure 8D), where the red color represents the time of occupancy and the solid white line the traveled distance.

To evaluate anxiety, each animal was allowed to freely explore the open field arena for 5 min at P30, monitoring the time spent in the center of the arena, a measure of the natural aversion of rodents to open spaces devoid of thigmotactic cues. Decreased time in the center of the arena is considered indicative of anxiety. As illustrated by Figure 8E, AS rats remained near the walls of the arena, avoiding the center and other areas devoid of thigmotactic cues, quantified as a significant decrease (*p* < 0.005, *N* = 12) in the total time spent in the center of the arena, compared to CS rats (*N* = 12). Only TNF-α+IFN-γ- MSC-S reversed the effect of PA (*p* < 0.005, *N* = 10).

Recognition memory was evaluated by the novel object recognition test at P30 with a 24 h delay between the first exposure and the recalling session. The time spent exploring a novel object was considered as a memory index (time spent exploring the new object divided by the total time spent exploring the old and new objects). Values of memory index lower than 0.5 indicate a recognition memory deficit [62]. AS rats showed a decreased memory index (by 50%, *p* < 0.005), compared to that shown by CS rats (Figure 8F). DFX-MSC-S (*p* < 0.05) or TNF-α+IFN-γ-MSC-S (*p* < 0.05) treatment prevented the decrease in memory index observed in AS rats, which improved up to the level observed in the CS group at P30 (Figure 8F).

## 3. Discussion

The model of global perinatal asphyxia developed by Bjelke and co-workers [63] over the past 25 years [21] closely mimics the relevant aspects of human delivery, a condition required for studying the early phase of PA, as observed in the clinical setup, acknowledging the fact that the model implies oxygen interruption, but not any additional invasive lesions, including, for example, vessel occlusion [64]. Moreover, it is followed by hypoxemia, acidosis, and hypercapnia, mandatory criteria for clinically relevant events of PA [65]. It is largely noninvasive; allows studying short- and long-term consequences of the insult in the same preparation, and it is highly reproducible among laboratories. Although the temporality has been criticized, arguing that the brain of neonate rats is premature when compared to the neonatal human brain, the statement mainly refers to the neocortex [66]. Our view is that the degree of maturity depends upon the tissue and functions selected for the comparisons, vulnerability relating to both the timing and the location of the insult [66,67].

Temperature control after brain injury is critical to optimizing recovery and functional outcome in neonatal rats [68,69]. Currently, therapeutic hypothermia is the only routinely used clinical intervention for full-term infants with PA. This intervention reduces death and disability, but it does not provide a complete protection [3].

The present study shows that intranasal administration to asphyxia exposed rat pups of secretome derived from MSC preconditioned with either DFX or TNF-α+IFN-γ: (i) fully suppressed the long-lasting hippocampal oxidative stress (elevated GSSG/GSH ratio) observed following PA at P7 and P60; (ii) induced NRF2 nuclear translocation and decreased NRF2 protein levels in cytoplasm at P7; (iii) increased the NQO1 antioxidant protein levels at P7; (iv) decreased nuclear NF-κB/p65 associated neuroinflammation and microglial reactivity at P7; (v) decreased cleaved-caspase-3 protein levels (a critical executioner of apoptosis) at P7; (v) improved behavioral development evaluated by the righting, negative geotaxis, and cliff aversion reflexes, motor coordination, locomotion, decreased anxiety, improved recognition memory deficits; (vi) MSC-S preconditioned with either DFX or TNF-α+IFN-γ, showed similar effectiveness to prevent the PA-induced deleterious effects. However, DFX- but not TNF-α+IFN-γ-preconditioned MSC-S increased NQO1 protein levels.

The broad spectrum of molecules secreted by MSC is known as conditioned medium or secretome. In humans, the MSC secretome consists of hundreds of biologically active molecules, including anti-inflammatory cytokines, trophic factors, and antioxidant molecules [70,71]. The paracrine potential of MSC can be boosted by in vitro preconditioning of these cells with environmental or pharmacological stimuli, enhancing their therapeutic efficacy. It has been demonstrated that MSC incubated under serum-deprived conditions, which mimic both, an ischemic and a hypoxic condition, increased the secretion of pro-survival and pro-angiogenic autocrine and paracrine substances, including vascular endothelial growth factor (VEGF)-A, angiopoietins, insulin growth factor (IGF)-1, and hepatic growth factor (HGF). This hypoxia-induced preconditioned environment has been linked to the induction of hypoxia-inducible factor (HIF)-1α, a transcription factor that binds to the hypoxia response elements in HIF-1α target genes, leading to hypoxia-preconditioning, protecting against subsequent energy deficits and/or brain ischemia [71]. In agreement with these findings, in a previous report we observed that the preconditioning of human AD-MSC, with 400 μM of the iron chelator DFX for 48 h, increased HIF-1α levels and upregulated the mRNA levels of pro-angiogenic factors such as VEGFα and angiopoietin 1, also increasing the expression of potent neuroprotective factors including nerve growth factor (NGF), glial cell-derived neurotrophic factor (GDNF), neurotrophin-3 (NT-3), cytokines with anti-inflammatory activity, such as IL-4 and IL-5, and antiapoptotic factors [53]. Moreover, in a model of diabetic polyneuropathy (DPN), a multifactorial disease, the secretome from MSC-DFX reverted DPN, improving thermal and mechanical sensitivity, restoring intraepidermal nerve fiber density, reducing neuronal and Schwann cell apoptosis, improving angiogenesis, and reducing oxidative stress and chronic inflammation of peripheral nerves, resulting in re-epithelialization of the injured skin, increasing blood vessels in the wound bed of a skin injury model mimicking foot ulcer [53,54]. The present study suggests that this therapeutic intervention is also effective to prevent the oxidative stress, neuroinflammation and cell death induced by PA.

The secretome from MSC preconditioned with 10 ng/mL TNF-α and 15 ng/mL IFN-γ for 40 h displayed enhanced antioxidant and anti-inflammatory activity, characterized by increased levels of anti-inflammatory cytokines IL-10 and TGF-β1, compared to the secretome derived from non-preconditioned MSC [58]. Moreover, TNF-α+IFN-γ-preconditioned MSC promoted immunosuppression and immunomodulation [47,55]. These results are in line with previous reports [58], indicating that pro-inflammatory preconditioning of MSC greatly improved the production and secretion of several anti-inflammatory factors, also participating in glutamate homeostasis.

Changes in the expression and function of glial-specific glutamate transporters have been demonstrated in a variety of models of brain insults and CNS pathologies, including drug abuse and chronic drug consumption, leading to oxidative stress and neuroinflammation, inhibiting the expression of the glutamate transporter (GLT)-1, and perpetuating drug intake [58]. These effects were reversed with intranasal administration of secretome derived from AD-MSC, preconditioned with TNF-α+IFN−γ, inhibiting chronic ethanol and nicotine self-administration [58]. Furthermore, these effects were abolished by GLT-1 knockdown, showing the important participation of GLT-1 on the secretome effect [58]. In PA, glutamate homeostasis by astrocytes appears to play a major role on the injury induced by PA [11,12,72]. It has been shown that chronic hypoxia reduces GLT-1 function and its expression in astrocytes of white matter [73] and striatum [74]. While in this study we focused mainly on MSC-S effects in hippocampus, the combined evidence suggests that glutamate metabolism may be prevented by TNF-α+IFN-γ-preconditioned MSC-S; decreasing the effect of altered glutamate metabolism induced by PA.

Thus, MSC-S pre-conditioned by proinflammatory cytokines, in addition to releasing anti-inflammatory and antioxidant factors, also regulates glutamate homeostasis by the GLT-1 transporter [58], while preconditioning of MSC by DFX increases HIF-1α, anti-inflammatory cytokines, and total antioxidant capacity [53]. Hence, the difference between DFX and TNF-α+IFN-γ-preconditioned MSC depends upon the molecules to be secreted by MSC [52]. In this study, both DFX and TNF-α+IFN-γ preconditioned MSC enhanced secretome efficiency, improving the outcome of PA, attenuating the pro-oxidative and inflammatory environment created by PA.

Intranasal administration of the secretome from multipotent stem/progenitor cells has been proposed as a strategy for treating other diseases, such as retinal ganglion cell loss [75,76], multiple sclerosis [77], Alzheimer’s disease [78], and Parkinson’s disease [79], including recently, alcohol and nicotine consumption [58]. In all these models, MSC-S has shown anti-inflammatory and antioxidant effects, also improving neurodevelopment and memory function. However, to our knowledge, no studies have previously reported on the use of secretomes as a therapeutic strategy for PA.

Intranasally delivered MSC-S products to CNS were shown by Ezquer et al. [80] to reach the brain within 6 h after administration, visualized as fluorescently labeled MSC-S exosomes in olfactory bulb, cerebral cortex, dorsal and ventral striatum, thalamus, hippocampus, and brainstem [80].

We previously reported that PA induced a regionally specific and sustained GSSG/GSH increase, decreasing catalase activity and tissue reducing capacity, observed up to P14 [7]. An increased GSSG/GSH ratio was also observed in the present study at P60, indicating prolonged oxidative stress months after the injury. This study further reports that the GSSG/GSH ratio was decreased by intranasal administration of DFX- or TNF-α+IFN-γ-MSC-S, an effect observed in hippocampus at both P7 and P60.

NRF2 activation is an important endogenous antioxidant mechanism protecting against oxidative stress [22,24,81,82]. It has been shown that under hypoxia-ischemia conditions, the expression of NRF2 is enhanced, released from Keap1, and translocated to the nucleus, binding to the ARE sequences. The NRF2-ARE system increases the expression of multiple cytoprotective antioxidant and glutathione generating enzymes, including NQO1 and HO1 [22,24]. In this study, cytoplasmic protein levels of NRF2 were upregulated by PA, compared to controls, although the NRF2 protein was not increased in the nucleus, suggesting that the endogenous upregulation of NRF2 was not sufficient to protect brain neurons and glial cells against PA injury, according to what has been found in a global ischemia model [83]. Moreover, it has been reported that oxidative stress can hamper Keap1-mediated NRF2 ubiquitination [84], an effect that might agree with increased NRF2 cytosolic levels found in the PA-exposed rats. Intranasal administration of DFX- or TNF-α+IFN-γ-MSC-S significantly elevated NRF2 levels in the nucleus, while simultaneously decreasing the levels in the cytoplasm of PA-exposed rats at P7, suggesting NRF2 pathway activation.

The antioxidant protein NQO1, downstream from NRF2, markedly increased in PA-exposed rats treated with DFX-MSC-S. However, the expression of other NRF2 effectors may occur earlier than P7. Therefore, no increase in HO1 protein was seen after PA and/or MSC-S treatment. It has been observed in a model of cerebral hemorrhage in rats that the expression of HO-1 is time-dependent, with a peak 3 days after the insult, then drastically dropping [85]. In the neonatal hypoxia-ischemia (HI) model by unilateral common carotid artery ligation at P7, treatment with δ opioid receptor agonists increased NQO1 over HO1 levels [86]. Moreover, argon treatment of HI rats enhanced NRF2, NQO1, and SOD1 levels [87]. The silencing of NRF2 decreased NQO1 levels, further supporting the relevance of NQO1 as a NRF2 effector following hypoxic conditions [88]. In agreement, this report shows that DFX-MSC-S increased NQO1 levels.

The NRF2 activation resulting from MSC treatment has been pointed out in other pathologies. For instance, cell-free conditioned media containing the secretome of stem cells added to cultured human alveolar type-1 epithelial cells, or administrated to adult rat lungs by tracheal instillation, ameliorated oxidative damage activating the NRF2 pathway [89]. Furthermore, MSC transplanted into a mouse model of liver failure improved the survival rate of the mice, also associated to NRF2 upregulation resulting in subsequent antioxidant activity [90], supporting the view that NRF2 might be the main actor in the mechanism by which MSC and MSC-S exert their therapeutic effects. Here, it was found that the secretome from MSC, preconditioned with pro-inflammatory or hypoxic-mimetic molecules, induced NRF2 nuclear translocation in the hippocampus, suggesting activation of the NRF2 pathway, in line with the remarkable increase of NQO1.

In the present PA model, MSC-S treatment decreased NF-κB/p65 nuclear translocation. This finding can be associated with NRF2 activation, since NRF2 and p65 may antagonize each other [91,92]. The antagonism exerted by NRF2 on p65 translocation has been described in cancer cells, showing that simultaneous activation of p65 and NRF2 antagonize each other by competing for the co-activator cAMP-response element-binding protein (CREB)-binding protein (CBP)-p300 complex, leading to inhibition of the NRF2 pathway [93]. Moreover, p65 decreases NRF2 binding to ARE sequences, increasing the NRF2-inhibitor Keap1 nuclear translocation, which releases NRF2 from the nucleus to the cytoplasm, where it is recruited by the ubiquitin ligase Cul3-Rbx E3 complex to be degraded by ubiquitination [91,94].

These findings are relevant, suggesting that MSC-S treatment increases NRF2 nuclear translocation in asphyxia-exposed rats, decreasing p65 nuclear translocation and, decreasing pro-inflammatory cytokine transcription, an issue that, however, remains to be further investigated.

The antagonism between NRF2 and NF-κB/p65 can occur in the CNS, where astrocytes are the main protagonists. In the absence of NRF2, astrocytes induce inflammation via p65 activation, leading to downstream pro-inflammatory cytokine transcription [95]. This could also imply microglia since their morphological changes decreasing the length of primary and secondary processes in *stratum radiatum* of CA1 hippocampus of asphyxia-exposed rats reflect an inflammatory phenotype. This phenotype was reverted by TNF-α+IFN-γ-MSC-S, as shown by this study. In the neonatal HI model at P10, human umbilical cord tissue-derived MSC administered intranasally induced beneficial effects observed in hippocampus, namely a decrease of neuronal loss and a reduction of neuroinflammation markers, including a decrease of activated microglia and astrocyte density [96]. Furthermore, in a transgenic Alzheimer´s disease mice model, the treatment with exosomes from hypoxic-preconditioned MSC decreased the activation of astrocytes and microglia, promoted the transformation of microglia to dendritic cells and down-regulated pro-inflammatory cytokines (TNF-α and IL-1β) in mice hippocampus [97], suggesting that the preconditioning of MSC with other hypoxia mimetic molecules, such as DFX could also prevent glial reactivity, an issue that remains to be further investigated.

Increased TUNEL-positive cell death and cleaved-caspase-3 have been described in hippocampus at P1, P3, and P7 in PA rats [5,7,25], even at P30 [6]. In this report, we found that treatment with either DFX-MSC-S or TNFα−IFNγ-MSC-S decreased PA-induced hippocampus cell death evaluated at P7, monitored by cleaved-caspase-3, suggesting decreased apoptosis.

The behavioral consequences of PA were evaluated by monitoring several developmental-dependent reflexes, deficits in motor coordination, locomotor activity, anxiety, and recognition memory [35,36,41], which were improved by MSC-S treatment. Intranasal administration of human umbilical cord tissue-derived MSC to neonatal HI induced by unilateral common carotid artery ligation at P10, decreased the behavioral impairments observed on negative geotaxis, evaluated four days after intranasal delivery, recovering the values up to those observed in control animals [96], similar to the present results obtained with DFX-MSC-S administered to asphyxia-exposed rats, observed at P14. In a mouse model of Alzheimer’s disease, it was shown that systemic intravenous delivery of human umbilical cord tissue-derived MSC improved memory function, along with reduction of amyloid plaques and gliosis, increasing neuronal density in cerebral cortex and hippocampus [78]. In agreement, we found here that MSC-S treatment improved the memory index evaluated by the novel object recognition test. The secretome from MSC injected into the substantia nigra and striatum of 6-hydroxydopamine-treated rats, a Parkinson’s disease rat model [79], increased the number of dopaminergic neurons and terminals in the injected hemisphere, reflected in the motor outcome, as evaluated by the rotarod test [79,98].

The time at which reflexes, such as righting, negative geotaxis, and gait are evaluated, is relevant, considering neonatal development, providing specific evaluation windows [59,60]. Righting reflex, cliff aversion, and negative geotaxis were evaluated in this study, finding significant delays in asphyxia-exposed animals, in agreement with previous reports [59,60]. At later time-points, no significant differences can be observed between control and asphyxia-exposed rats. As previously reported, asphyxia led to deficits in nonspatial memory, assessed by the novel object recognition test [35], anxiety-related behavior [99], and altered emotional behavior [39]. Nevertheless, there are studies reporting no changes in gross motor activity [35], or open-field behavior [99]. MSC-S treatment prevents the developmental deficits elicited by PA (righting, cliff aversion, and negative geotaxis reflexes), and on locomotion and motor coordination, as well as on balance, novel object recognition memory, and anxiety. These findings suggest a sustained effect of secretome of MSC preconditioned with either DFX or TNF-α+IFN-γ, also impacting on motor development and memory.

## 4. Materials and Methods

### 4.1. Animals

Wistar rats from the animal station of the Molecular & Clinical Pharmacology Programme, ICBM, Medical Faculty, University of Chile, Santiago, Chile, were used for all experiments. The animals were kept in a temperature- and humidity-controlled environment with a 12/12 h light/dark cycle, with access to water and food ad libitum, permanently monitoring their well-being by qualified personnel.

### 4.2. Ethics Statement

Human MSCs were isolated from subcutaneous adipose tissue (abdominal liposuction), after written informed consent. All protocols were approved by the Ethics Committee of the Medical Faculty, Clínica Alemana-Universidad del Desarrollo (Protocol 2015-40). All procedures were conducted in accordance with the animal care and use protocols established by a Local Ethics Committee for experimentation with laboratory animals at the Medical Faculty, University of Chile (Protocol CBA#1058 FMUCH, 19255-MED-UCH, date of approval: 06 May 2019) and by an ad-hoc commission of the Chilean Council for Science and Technology Research (CONICYT) (FONDECYT #1190562, date of approval 4 March 2019), endorsing the principles of laboratory animal care. Animals were permanently monitored (on a 24 h basis) regarding their wellbeing, following the ARRIVE guidelines for reporting animal studies (www.nc3rs.org.uk/ARRIVE).

### 4.3. Perinatal Asphyxia

Pregnant Wistar rats in the last day of gestation (G22) were euthanized and hysterectomized. Three or four pups per dam were removed immediately to be used as non-asphyxiated cesarean-delivered controls. The remaining fetus-containing uterine horns were immersed in a water bath at 37 °C for 21 min, to maintain uterine thermal conditions [100]. Following asphyxia, uterine horns were incised and pups removed, stimulated to breathe by wiping the amniotic fluid and by tactile stimulation of the oral region with pieces of medical wipes as well as by pressing the thorax. This resuscitation maneuver implies expert and skillful handling, taking a long time (4–6 min) to stimulate the first gasp, and an even longer time to establish regular breathing, always supported by gasping. Then, after a 40 min observation period on a warming pad within the thermal neutral ranges 36–37 °C [100], the pups were evaluated with an Apgar scale adapted for rats, according to Dell’Anna et al. [101], randomly assigned to each experimental condition and nursed by a surrogate dam.

### 4.4. Isolation, Expansion, and Characterization of Human Adipose Tissue-Derived MSC-S

Human adipose tissue-derived mesenchymal stem cells (hAD-MSC) were isolated from fresh subcutaneous adipose tissue samples obtained from liposuction aspirates of patients undergoing cosmetic liposuction after obtaining written informed consent at the Clínica Alemana, Santiago, Chile, as previously described [53]. All protocols were approved by the Ethics Committee of the Medical Faculty, Clínica Alemana-Universidad del Desarrollo. After two subcultures, cells were characterized according to adipogenic and osteogenic differentiation potential and by the expression of putative surface markers as described previously [53].

### 4.5. Preconditioning of Human Adipose Yissue-Derived MSC and Secretome Generation

Human AD-MSC (passage 3) at 70% of confluence were preconditioned by incubation in minimal essential medium (α-MEM, Gibco, Grand Island, NY, USA) supplemented with 10% fetal bovine serum (FBS; HyClone, South Logan, UT, USA), and 0.16 mg/mL gentamicin (Sanderson Laboratory, Chile) at 37 °C and 5% CO_2_ plus (i) 400 µM deferoxamine (DFX, Sigma-Aldrich, USA) in α-MEM without FBS for 48 h, as described previously [54] to improve the production of antioxidant, anti-inflammatory, and angiogenic factors [53]; or plus (ii) 10 ng/mL TNF-α and 15 ng/mL IFN-γ (R&D System, Minneapolis, MN, USA) according to previous reports [58,80], to improve the production of anti-inflammatory factors.

After preconditioning, the cells were washed three times with phosphate-buffered saline (PBS) and incubated for 48 h with α-MEM. After that, the culture medium (secretome) was collected and centrifuged at 400× *g* for 10 min to remove the whole cells. The supernatant was centrifuged again at 5000× *g* for 10 min to remove cell debris. This process reduces the contamination of the secretome with proteins released by the rupture of the cells. Finally, the secretome was filtered in 0.22 μm filters and concentrated 50 times (v/v) by 3 kDa cutoff filters (Millipore, Carrigtwohill, Ireland). The protein concentration was determined by the BCA protein assay kit (Thermo Scientific, Waltham, MA, USA), and the secretome was frozen at −80 °C until use.

### 4.6. MSC-S Intranasal Administration

Intranasal administration was performed according to a previous report [85]. Briefly, a total volume of 16 µl, containing 6 μg of secretome proteins (derived from 2 × 10^5^ preconditioned-MSC), was administrated intranasally 2 h post-asphyxia or in two doses, 2 h and 7 days post-asphyxia, as follows: every 5 min, one μL of solution was administered intranasally by a small cannula into alternative sides of the nasal cavity (eight times in each nostril). As a control 16 μL of saline (0.9% NaCl) was administered as above. Rat neonates were evaluated at P7, and at different postnatal periods up to P60.

### 4.7. Tissue Sampling for Biochemical Protein Analysis

For tissue sampling, rats (females and males) were euthanized at P7 and P60. The brain was quickly removed to dissect out hippocampus for biochemical assays. The procedure was performed on ice, using a newborn rat brain slicer (Zivic Instruments, Pittsburgh, PA 15237, USA), frozen in liquid nitrogen, and kept at −80 °C until use.

### 4.8. GSSG/GSH Ratio Determination

Rats were anesthetized and perfused intracardially with 100 mL of 0.1 M PBS (pH 7.4). Hippocampi were extracted and mixed with three volumes of ice-cold potassium buffer containing 5 mM EDTA, pH 7.4, and flash-frozen and stored at 80 °C until homogenization. Reduced glutathione (GSH) and oxidized glutathione disulfide (GSSG) content was determined as described previously [7,55]. Briefly, in the assay for total GSH and GSSG, GSSG in the sample was first converted into GSH with glutathione reductase and NADPH. The total free thiol group of GSH was reacted with the sulfhydryl reagent DTNB (5,50 -dithiobis-2- nitrobenzoic acid) yielding a product that absorbs at 412 nm. GSSG per se in the homogenate was measured by adding 2-vinyl pyridine, a thiol scavenger that traps GSH, preventing GSH from binding to DNTB. The excess of 2-vinyl pyridine was neutralized with triethanolamine. Thereafter, GSSG was converted into GSH by glutathione reductase (Sigma-Aldrich cat.# G3664, MO, USA) and NADPH (Sigma-Aldrich cat. N1630, MO, USA), DNTB (Sigma-Aldrich, cat. D-8130, MO, USA) was added and absorbance measured at 412 nm.

### 4.9. Protein Extraction

Cytoplasmic and nuclear protein extracts were obtained using the Nuclear extraction kit (Cayman Chemical, cat. 10009277, Ann Arbor, MI, USA) according to the manufacturer’s instructions. Hippocampi at P7 were homogenized in 200 µl of provided cytosolic extraction buffer at 4 °C. Later, 50 µl of nuclear extraction buffer was used to obtain nuclear proteins. Protein concentration was determined using a bicinchoninic acid (BCA) assay kit (Pierce, Thermo Scientific, Rockford, IL, USA). Absorbance was measured at 562 nm in a 96 well plate Reader (Synergy HT Biotek Instruments, Inc., Winooski, VT, USA).

### 4.10. Western Blots

Protein electrophoresis was performed using SDS polyacrylamide gel, at 6% for stacking and 12% as separating gel. 50 µg of cytoplasmic protein samples and the same volume of corresponding nuclear extract were charged in each gel well. Protein samples were mixed with loading buffer (30% glycerol, 6% SDS, 15% DTT, 0.2% bromophenol blue, and 120 mM Tris HCl buffer pH 6.8) and then kept at 95 °C for 5 min. Electrophoresis was performed at constant 25 V overnight. Transfer to nitrocellulose membrane (Pall Corporation, Pensacola, FL, USA) was performed at constant 300 mA for 90 min at 4 °C. Protein transfer to membrane was evaluated using Ponceau red staining (0.1% Ponceau red, 0.1% acetic acid). Ponceau was removed with Tris-buffered saline (TBS) and then membranes were blotted with 5% non-fat milk (Bio-Rad, cat. 1706404, Hercules, CA, USA) diluted in TBS/ 0.1 % Tween-20 (TBS-T), at room temperature for 90 min. Primary antibodies were incubated overnight at 4 °C, at the following dilutions in TBS-T: (1:1000) anti-NRF2 (Abcam, cat. ab89443, monoclonal mouse, Cambridge, MA, USA); (1:1000) HO-1 (Cell Signaling Technology (E3F4S), cat. 43966S, monoclonal rabbit, Danvers, MA, USA); (1:500) NQO1 (Cell signaling Technology (A180), cat. 3187S, monoclonal mouse, Danvers, MA, USA); (1:100) p65 (Santa Cruz Biotechnology, sc-372, polyclonal rabbit, CA, USA); (1:500) caspase-3 (Cell Signaling Technology (8G10), cat. 9665S, monoclonal rabbit, Danvers, MA, USA); (1:1000) β-tubulin (Cell Signaling Technology, cat. 2146S, polyclonal rabbit, Danvers, MA, USA), and (1:750) histone H-4 (Cell Signaling Technology (D2X4V), cat. 13919S, monoclonal rabbit, Danvers, MA, USA). Thereafter, membranes were washed with TBS-T three times for 10 min at room temperature with constant shaking and incubated with the corresponding horseradish peroxidase (HRP)-conjugated secondary antibody (Thermo Scientific Pierce, Rockford, IL, USA) diluted 1:10,000 in TBS-T, for 1 h at room temperature with constant shaking. Then, membranes were washed in TBS-T three times for 10 min at room temperature under constant shaking. Membranes were developed using the Pierce^TM^ Enhanced chemiluminescence Western Blotting Detection kit for HRP (Thermo Fisher Scientific, Inc., cat. 32209, Rockford, IL, USA). Finally, a ChemiScope 3400 (ClinX Sciences Instruments Co, Ltd., Shanghai, China) was used to capture the chemiluminescence. Images were processed by Image J 1.52f software (National Institutes of Health, USA). The area values were normalized to their respective β-tubulin or histone H-4 mark for cytoplasmic and nuclear extracts, respectively.

### 4.11. Microglia Reactivity

Immunofluorescence (IF) against microglial ionized calcium binding adaptor molecule 1 (IBA-1) and nuclei counterstained with 4,6-diamino-2-phenylindol (DAPI) was evaluated in coronal hippocampal cryo-sections as previously described [50,102]. Briefly, controls and asphyxia-exposed rats at P7 were anesthetised and perfused intracardially with 0.1 M PBS (pH 7.4), followed by a formalin solution (4% paraformaldehyde in 0.1 M PBS, pH 7.4). The brain was removed from the skull, post-fixed in the same fixative solution overnight at 4 °C, and immersed in 10% sucrose containing 0.1 M PBS for 48 h and, subsequently, in 30% sucrose at 4 °C for 48 h. Coronal sections (30 μm thick) of the hippocampi (between Bregma −1.40 and −2.00 mm, https://www.ial-developmental-neurobiology.com/en/publications/collection-of-atlases-of-the-rat-brain-in-stereotaxic-coordinates) were obtained using a cryostat (Thermo Scientific Microm HM 525, Germany). Sections were rinsed with 0.1 M PBS and treated with blocking solutions for 1 h, incubated with primary antibody against IBA-1 (rabbit, Wako Chemicals USA, cat. 019-19741; 1:500 in blocking solution containing 1% BSA, 10% NGS, and 0.3% Triton X 100) overnight at 4 °C. After repeated rinsing with 0.1 M PBS, sections were incubated with the secondary antibody anti-rabbit-Alexa 488 (goat, Thermo Fisher Scientific, IL, USA, cat.A-11034; 1:500 incubated in blocking solution containing 1% NGS) and counterstained with DAPI (Thermo Fisher Scientific, cat.D1306, 0.02 M; 0.0125 mg/mL; for nuclear labelling) for 2 h. After rinsing, the samples were mounted on silanized slides with Fluoromount-G^TM^ (Thermo Fisher Scientific, IL, USA) and examined by confocal microscopy (Olympus-fv10i, Center Valley, PA, USA).

### 4.12. Microscopy and Image Analysis of IBA-1 Positive Cells

Microphotographs (5–6) were taken from the *stratum radiatum* of the CA1 hippocampal region in the field of an Olympus FV10i microscope (Center Valley, PA, USA), using a 60× objective lens (NA1.30). Images were captured using FV10-ASW-2b software (Olympus). The area inspected for each stack was 0.04 mm^2^ and the thickness (Z-axis) was measured for each case. The total lengths of primary and secondary processes for six IBA-1 positive cells per Z-stack were analyzed by three-dimensional reconstruction using the Simple Neurite Tracer plugin of FIJI image analysis (http://fiji.sc/Simple_Neurite_Tracer) [103] following the protocol described by Tavares et al. [104].

### 4.13. Evaluation of Behavioral Parameters

For behavioral parameters, rats (females and males) were evaluated at P1, P7, P14, P30, and P60. For the righting reflex, rats were placed in the supine position and the time to turn over to the prone position was determined, ensuring that the four paws were in contact with the standing surface (evaluated at P1) [61]. For evaluation of cliff aversion, the rat was placed with its forelimbs and head on the edge of a surface raised up to 30 cm for 30 s. The time at which the rat rotated to avoid falling was recorded at P14. For evaluation of negative geotaxis, the rat was placed face down on a 30° inclined grid of 30 × 20 cm. The hindlimbs of the rat were placed in the middle of the grid. The time at which the rat turned to face up the slope and climb up the grid with its forelimbs reaching the upper rim was recorded at P14.

Motor coordination was evaluated with a rotarod at P60, with a rod axis accelerated from 5 to 40 rpm over a five-minute period, as previously described [102].

Exploratory behavior was monitored at P30 with a video camera, with the animals placed in a square arena (open field) in a noise-free room as previously described [38,62]. When monitoring rats at P7, the rat was placed in the center of a square arena (21 × 29 cm, with a continuous wall 15 cm high) to freely explore for 5 min, monitoring the total traveled distance (cm) (locomotor activity). The total time spent in the central zone of the open-field (s) was also monitored. A decreased time in the center of the arena is considered as a measure of anxiety [105], exploiting the natural aversion of rodents to open spaces devoid of thigmotactic cues. This measure was recorded with an automated video tracking Matlab routine (Mathworks, Inc, Natick, MA, USA).

The novel object recognition test was used to evaluate recognition memory, assessed at P30 as previously described [38,62]. Briefly, the animals were first habituated to a box and then exposed to two identical plastic objects for 5 min, letting the animals freely explore and get familiar with the objects. After 24 h, each animal was tested for object recognition memory by placing the animal in the box with one of the previously exposed objects, and a new one for 5 min. Offline analysis of the video recording evaluated the time expended exploring the new object, versus the total time exploring the old and the new object, as an index of memory recognition [106]. A memory index below 0.5 indicated memory recognition deficit [62,106]. This index was recorded with an automated video-tracking Matlab routine (Mathworks, Inc, USA).

### 4.14. Statistical Analysis

All data are expressed as means ± SEM. One-way ANOVA for multiple comparisons were performed, followed by Benjamini, Hochberg, and Yekutieli as post hoc tests. For behavioral analysis, nonparametric Kruskal-Wallis one-way ANOVA on rank was performed, followed by Benjamini, Hochberg, and Yekutieli post hoc tests. To compare continuous variables between two groups, the Mann Whitney test was used. Statistical analysis was conducted using GraphPad Prism 5 (GraphPad Software, Inc, San Diego, CA, USA) software, setting a level of *p* < 0.05 for statistical significance. To facilitate text reading ANOVA data are presented in the Legends to Figures.

## 5. Conclusions

In conclusion, intranasal administration of preconditioned MSC-derived secretome to PA-exposed rats decreased oxidative stress, neuroinflammation, and cell death, improving the neurobehavioral development, motor coordination, locomotor activity, recognition memory, and decreasing anxiety. The secretome yielded similar effects whether the MSC were preconditioned with either DFX or TNF-α-+IFN-γ, improving the PA outcome. Thus, intranasal administration of preconditioned MSC-S is a novel therapeutic strategy to prevent the short- and long-term effects of perinatal asphyxia.

## Figures and Tables

**Figure 1 ijms-21-07800-f001:**
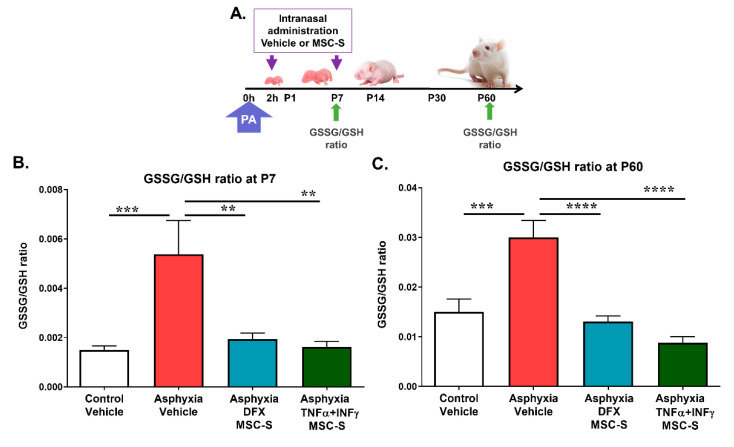
Effect of intranasal administration of DFX- or TNF-α+IFN-γ-MSC-S on hippocampal oxidative stress (GSSG/GSH ratio) induced by PA evaluated at P7 and P60. (**A**). Schematic representation of intranasal administration of vehicle or MSC-S monitoring schedules. Animals evaluated at P7 received only one MSC-S dose (2 h after birth), and those evaluated at P60 received two doses (2 h after birth and at P7). The GSSG/GSH ratio was determined in the hippocampus of control and asphyxia-exposed rats (intranasally treated with vehicle or either DFX- or TNF-α+IFN-γ-MSC-S, evaluated at P7 and P60. (**B**). GSSG/GSH ratio at P7: PA-exposed rats treated with vehicle (AS) (red bar) showed an increased GSSG/GSH ratio, versus vehicle-treated (CS) controls (white bar). Administration of a single dose of DFX-MSC-S (cyan bar) or TNF-α+IFN-γ-MSC-S (green bar) fully reversed the effect of PA on the GSSG/GSH ratio. Data represent means ± SEM. One-way ANOVA, F_(3,30)_ = 6.375, *** p* < 0.005. Benjamini, Hochberg, and Yekutieli post-hoc: **** p* < 0.0005; CS v/s AS, *** p* < 0.005; AS v/s Asphyxia-DFX-MSC-S, *** p* < 0.005; AS v/s Asphyxia-TNF-α+IFN-γ-MSC-S, *N* = 6–10 samples per group. (**C**). GSSG/GSH ratio at P60: AS rats (red bar) still displayed an increased GSSG/GSH ratio versus CS (white bar). Administration of two intranasal doses of DFX- (cyan bar) or TNF-α+IFN-γ-MSC-S (green bar) fully reversed the effect of PA on the GSSG/GSH ratio. One-way ANOVA, F_(3,16)_ = 18.55, ***** p* < 0.0001. Benjamini, Hochberg, and Yekutieli post-hoc: **** p* < 0.0005, CS v/s AS, **** *p* < 0.0001; AS v/s Asphyxia-DFX-MSC-S, ***** p* < 0.0001; AS v/s Asphyxia-TNF-α +IFN-γ-MSC-S, *N* = 4–6 samples per group.

**Figure 2 ijms-21-07800-f002:**
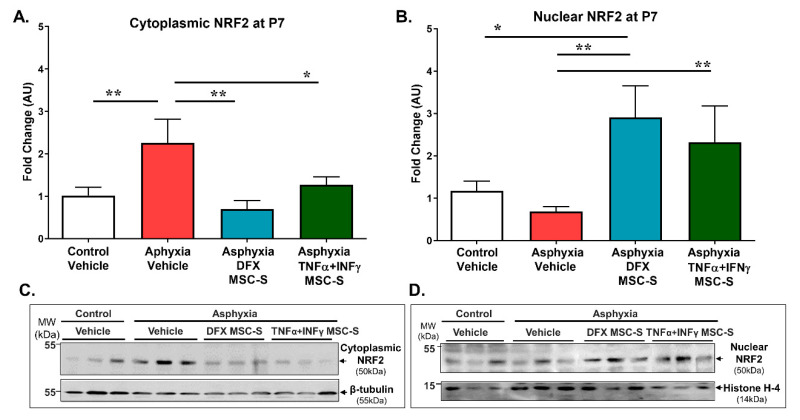
Effect of intranasal administration of DFX- or TNF-α+IFN-γ-MSC-S on nuclear and cytoplasmic NRF2 protein levels in hippocampus of PA-exposed and control rats at P7. (**A**) Cytoplasmic NRF2 protein levels; (**B**) Nuclear NRF2 protein levels. Hippocampal protein levels were evaluated by Western blots in control (white bar) and asphyxia-exposed rats (red bar), intranasally treated with vehicle- or DFX-MSC-S (cyan bar)- or TNF-α+IFN-γ-MSC-S (green bar)-MSC-S at P7. β-tubulin and histone H-4 were used as normalizers for cytoplasmic and nuclear extracts, respectively. Representative Western blot images are shown (**C**) and (**D**). Data are presented as ratio of protein/normalizer level, relative to vehicle controls. (**A**) PA-exposed rats treated with vehicle (AS) (red bar) showed an increased cytoplasmic NRF2 protein levels versus vehicle-treated (CS) controls (white bar). Administration of a single dose of DFX-MSC-S (cyan bar) or TNF-α+IFN-γ-MSC-S (green bar) reversed the effect of PA on cytoplasmic NRF2 levels. Data represent means ± SEM. One-way ANOVA, F_(3,23)_ = 3.973, ** *p* < 0.05. Benjamini, Hochberg, and Yekutieli post-hoc: ** p* < 0.05, CS v/s AS, *** p* < 0.005; AS v/s Asphyxia DFX-MSC-S; * *p* < 0.05; AS v/s Asphyxia-TNF-α+IFN-γ-MSC-S, *N* = 6–7 samples per group. (**B**). A single intranasal administration of DFX-MSC-S (cyan bar) or TNF-α+IFN-γ-MSC-S (green bar) to PA-exposed rats increased nuclear NRF2 levels, compared to AS (red bar) and to CS (white bar) rats. One-way ANOVA F_(3,30)_ = 3.772,* *p* < 0.05. Benjamini, Hochberg, and Yekutieli post-hoc: * *p* < 0.05, CS v/s Asphyxia-DFX-MSC-S; ** *p* < 0.005, AS v/s Asphyxia-DFX-MSC-S; ** p* < 0.05, AS v/s Asphyxia-(TNF-α+IFN-γ)-MSC-S, *N* = 8–10 samples per group.

**Figure 3 ijms-21-07800-f003:**
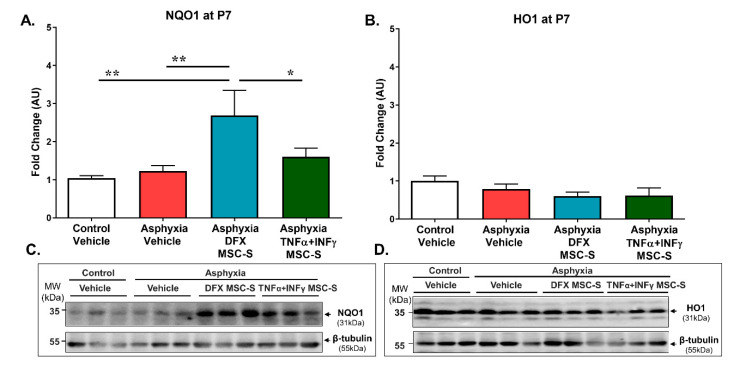
Effect of intranasal administration of DFX- or TNF-α+IFN-γ-MSC-S on NRF2 effector NQO1 and HO1 protein levels determined in hippocampus of PA-exposed and control rats at P7. (**A**) NQO1 and (**B**). HO1 hippocampal protein levels evaluated by Western blot in control (CS, white bar) and asphyxia-exposed rats (AS, red bar), intranasally treated with vehicle- or DFX -MSC-S(cyan bar) or TNF-α+IFN-γ-MSC-S (green bar) at P7. β-tubulin was used as normalizer for protein extracts. Data are presented as ratios of protein levels/normalizer levels, relative to vehicle control levels. Representative Western blot images are shown in (**C**) and (**D**). Data represent means ± SEM. (**A**) NQO1levels: A single intranasal administration of DFX-MSC-S to PA-exposed rats (cyan bar) increased NQO1 levels compared to AS (red bar) and CS rats (white bar). One-way ANOVA, F_(3,19)_ = 4.966,* *p* < 0.01 Benjamini, Hochberg, and Yekutieli post-hoc: *** p* < 0.005, CS v/s Asphyxia-DFX-MSC-S, *** p* < 0.005; AS v/s Asphyxia-DFX-MSC-S, * *p* = 0.05; Asphyxia-DFX-MSC-S v/s Asphyxia-TNFα+IFN-γ-MSC-S, *N* = 5–6 samples per group. (**B**) HO1 levels: No effect of PA and/or MSC-S treatment was observed on HO1 protein levels, evaluated in hippocampus at P7. One-way ANOVA, F_(3,29_ = 1.726, n.s. *N* = 8 samples per group.

**Figure 4 ijms-21-07800-f004:**
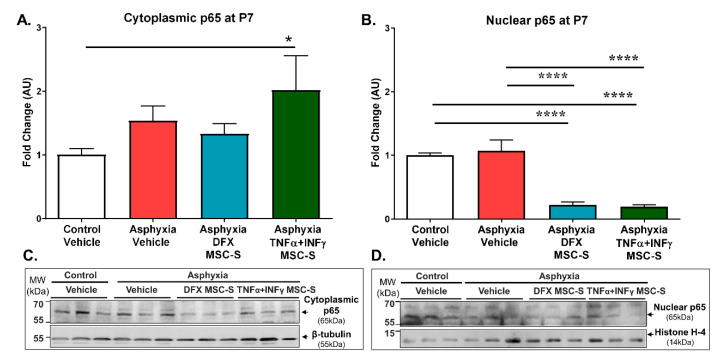
Effect of intranasal administration of DFX- or TNF-α+IFN-γ-MSC-S on cytoplasmic and nuclear p65 protein levels determined in hippocampus of PA-exposed rats at P7. (**A**) Cytoplasmic and (**B**). Nuclear p65 protein levels evaluated by Western blot at P7 in control (white bar) and asphyxia-exposed rats (red bar), intranasally treated with vehicle- or DFX-MSC-S (cyan bar) or TNF-α+IFN- γ-MSC-S (green bar). β-tubulin and histone H-4 were used as normalizers for cytoplasmic and nuclear extracts, respectively. Data are presented as ratios of protein levels/normalizer levels, relative to control levels. Representative Western blot images are shown in (**C**). and (**D**). Data represent means ± SEM. (**A**) Cytoplasmic p65 protein levels: An increased cytoplasmic p65 level was observed in PA treated with TNF-α+IFN-γ-MSC-S (green bar), compared to vehicle-treated control rats (CS, white bar). One-way ANOVA, F_(3,20)_ = 1.892, *p* = 0.164. Benjamini, Hochberg, and Yekutieli post-hoc: ** p* < 0.05, CS v/s TNF-α+IFN-γ-MSC-S, *N* = 5–6 samples per group. (**B)**. Nuclear p65 protein levels: A decrease in nuclear p65 levels was observed in PA-exposed rats treated with DFX- (cyan bar) or TNF-α+IFN-γ-MSC-S rats (green bar), compared to PA-exposed (AS) (red bar) and CS rats (white bar). One-way ANOVA, F_(3,18)_ = 23.99, ***** p* < 0.0001. Benjamini, Hochberg, and Yekutieli post-hoc: ***** p* < 0.0001, CS v/s Asphyxia-DFX-MSC-S; CS v/s Asphyxia-TNF-α+IFN-γ-MSC-S; AS v/s Asphyxia-DFX-MSC-S; AS v/s. TNF-α+IFN-γ-MSC-S, *N* = 4–6 samples per group.

**Figure 5 ijms-21-07800-f005:**
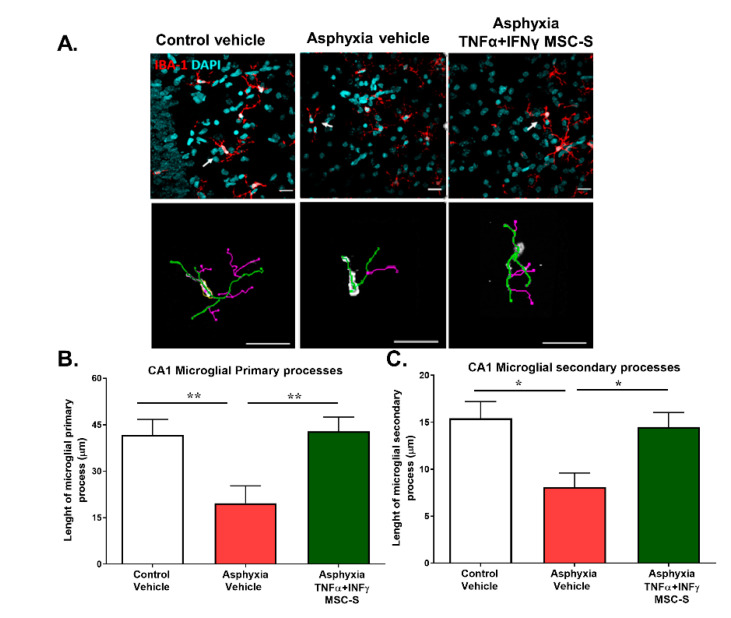
Effect of intranasal administration of TNF-α+IFN-γ-MSC-S on microglial process length in hippocampal CA1 region from PA-exposed rats at P7. (**A**) Representative microphotographs of immunofluorescence of IBA-1 (red) counterstained with DAPI (cyan) markers from controls and asphyxia rats treated with vehicle (AS) and asphyxia rats treated with-TNF-α+IFN-γ-MSC-S in the *stratum radiatum* of the CA1 hippocampal region at P7. White arrows indicate representative microglial cells depicted in the lower panel showing the three-dimensional reconstruction of their primary (green) and secondary (purple) processes, analyzed by FIJI software. (Bar: 20µm). Length of primary microglial processes. (**B**) Length of secondary microglial processes (IBA-1^+^ cells). (**C**) Data represent mean ± SEM. * *p* ˂ 0.05 ANOVA and Benjamini, Hochberg, and Yekutieli post-test. (**B**) Primary processes; ** *p* < 0.005, Control vehicle (CS) v/s AS; ** p* < 0.05, AS v/s Asphyxia- TNF-α+IFN-γ-MSC-S. (**C**). Secondary processes; *** p* < 0.005, CS v/s AS; * *p* < 0.05, AS v/s Asphyxia-TNF-α+IFN-γ-MSC-S, *N* = 5–8 samples per group.

**Figure 6 ijms-21-07800-f006:**
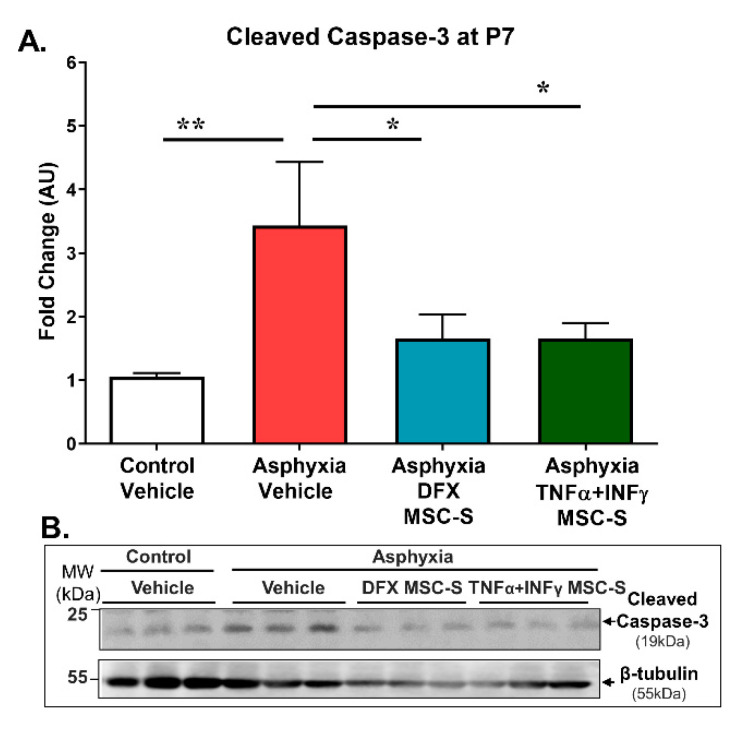
Effect of intranasal administration of DFX- or TNF-α+IFN-γ-MSC-S on hippocampal cell death (cleaved-caspase-3) induced by PA at P7. Cleaved-caspase-3 protein level evaluated by Western blot at P7 in control (CS, white bar) and asphyxia-exposed rats (AS, red bar), intranasally treated with vehicle- or DFX-MSC-S (cyan bar), or TNF-α+IFN-γ-MSC-S (green bar). β-tubulin was used as normalizer. Data are presented as ratios of protein levels/normalizer levels, relative to vehicle control levels. Data represent means ± SEM. (**A**) Cleaved-caspase-3 protein levels: AS rats showed an increased level of cleaved-caspase-3 protein, versus CS rats. Administration of a single dose of DFX- or TNF-α+IFN-γ-MSC-S reversed the effect of PA on cleaved-caspase-3 levels. One-way ANOVA, F _(3,22)_ = 3.985, ** p* < 0.05. Benjamini, Hochberg, and Yekutieli post-hoc: *** p* < 0.005; CS v/s AS; * *p* < 0.05; AS v/s Asphyxia-DFX-MSC-S; ** p* < 0.05; AS v/s Asphyxia-TNF-α+IFN-γ-MSC-S, *N* = 7–6 samples per group. (**B**) Representative Western blot image is shown.

**Figure 7 ijms-21-07800-f007:**
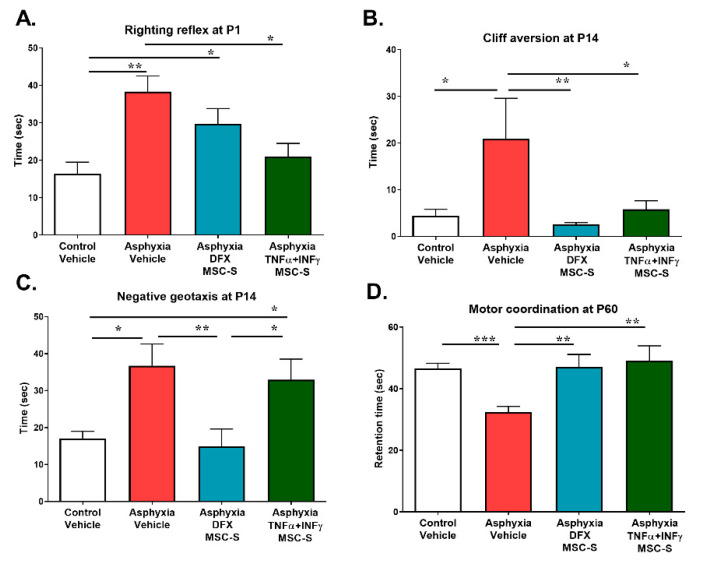
Effect of intranasal administration of DFX- or TNF-α+IFN-γ-MSC-S on the behavioral outcome induced by PA. Schematic representation of intranasal administration of vehicle or MSC-S and evaluation schedule are shown in Appendix A. (**A**) Righting reflex: rats were placed in a supine position and the time elapsed in seconds to turn over to a prone position, standing on all four paws, was recorded for control (CS, white bar) and asphyxia-exposed rats (AS, red bar) at P1, intranasally treated with vehicle, DFX-MSC-S (cyan bar), or TNF-α+IFN-γ-MSC-S (green bar). Data represent means ± SEM. Kruskal-Wallis test, *** p* < 0.05, Benjamini, Hochberg, and Yekutieli post-hoc: *** p* < 0.005; CS v/s AS, * *p* < 0.05; AS v/s Asphyxia- (TNF-α+IFN-**γ**)-MSC-S, ** p* < 0.05; CS v/s asphyxia-DFX-MSC-S, ** p* < 0.05; AS v/s Asphyxia-TNFα+IFN-γ-MSC-S; N = 38–35 samples per group. (**B**) Cliff aversion: the rat was placed with its forelimbs and head on the edge of a raised surface 30 cm from the floor for 30 s. The time at which the rat turned away from the cliff and moved its paws and head away from the edge was recorded at P14. Data represent means ± SEM. Kruskal-Wallis test, * *p* < 0.05, using Benjamini, Hochberg, and Yekutieli as post-hoc test: ** p* < 0.05; CS (white bar) v/s AS (red bar), *** p* < 0.005; AS v/s Asphyxia-DFX-MSC-S (cyan bar), ** p* < 0.05; AS v/s Asphyxia-TNFα+IFN-γ-MSC-S (green bar), *N* = 7–9 samples per group. (**C**) Negative geotaxis: the rat was placed facing down on a slope (30°). The hindlimbs of the pups were placed in the middle of the slope. The time in which the rat turned to face up the slope and climbed up with its forelimbs reaching the upper rim was recorded at P14. Data represent means ± SEM. Kruskal-Wallis test ** *p* < 0.005, followed by Benjamini and Hochberg, and Yekutieli post-hoc: * *p* < 0.05, CS (white bar), * *p* < 0.05; CS v/s Asphyxia-TNFα+IFN-γ-MSC-S (green bar), ** *p* < 0.005; AS v/s Asphyxia-DFX-MSC-S (cyan bar), ** *p* < 0.005; Asphyxia-DFX-MSC-S v/s Asphyxia-TNFα+IFN-γ-MSC-S, *N* = 5–8 per group. (**D**) Motor coordination: At P60, the rat was placed on a rotarod apparatus perpendicular to a rod axis, with its head facing in the opposite direction to the rotation around the rod’s axis. Trials ended once the rat had fallen and the time spent on the rod was recorded (retention time). Data represent means ± SEM. Kruskal-Wallis test, *** *p* <0.0005, followed by Benjamini, Hochberg, and Yekutieli post-hoc: *** *p* < 0.0005, CS (white bar) v/s AS (red bar), ** *p* < 0.005; AS v/s Asphyxia-DFX-MSC-S (cyan bar), ** *p* < 0.005; AS v/s Asphyxia-TNFα+IFN-γ-MSC-S (green bar), *N* = 19–23 samples per group.

**Figure 8 ijms-21-07800-f008:**
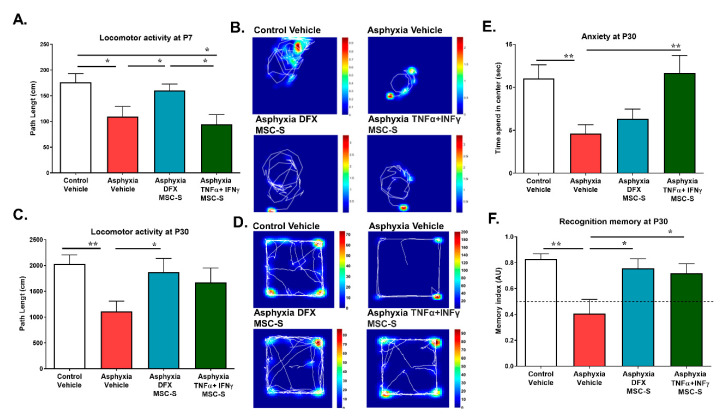
Effect of intranasal administration of DFX- or TNF-α+IFN-γ-MSC-S on locomotor activity, anxiety, and recognition memory. Locomotor activity: exploratory behavior was monitored in a square open field arena at P7 and P30. (**A**) Locomotor activity at P7. Data represent means ± SEM. Kruskal-Wallis, * *p* < 0.05, followed by Benjamini, Hochberg, and Yekutieli post-hoc: * *p* < 0.05, Control vehicle (CS, white bar) v/s Asphyxia vehicle (AS, red bar), * *p* < 0.05; CS v/s Asphyxia-TNFα+IFN-γ-MSC-S (green bar), ** p* < 0.05; AS v/s Asphyxia-DFX-MSC-S (cyan bar), ** p* < 0.05; Asphyxia-DFX-MSC-S v/s Asphyxia-TNFα+IFN-γ-MSC-S, N = 8–10 samples per group. (**C**) Similar effects on locomotor activity were observed at P30: Kruskal-Wallis, ** p* < 0.05, followed by Benjamini, Hochberg, and Yekutieli post-hoc: * *p* < 0.005, CS v/s AS; ** p* < 0.05, AS v/s Asphyxia-DFX-MSC-S), *N* = 8–12 samples per group. Representative tracking traces showing the occupancy maps of different groups of rats in the arena at P7: (**B**); and at P30: (**D**) The color bars represent the time of occupancy and the solid line the trajectory travelled. (**E**) Anxiety at P30. Quantification of the time spent by the rats at the center of the arena was monitored during a first 5 min observation period. AS rats remained close to the wall of the arena, avoiding the center and other areas devoid of thigmotactic cues. This behavior was reversed by TNFα+IFN-γ-MSC-S administration. Data represent means ± SEM. Kruskal-Wallis test, *** p* < 0.005, followed by Benjamini, Hochberg, and Yekutieli post-hoc: *** p* < 0.005; CS v/s AS, ** *p* < 0.005; AS v/s Asphyxia-TNFα+IFN-γ-MSC-S, *N* = 8–12 samples per group. (**F**) Novel object recognition memory at P30. First, the rat was exposed to two identical objects for 5 min. 24 h later, the rat was exposed to one of the same objects and to a new object. Memory index refers to the time spent exploring the new object divided by the total time spent exploring the old and the new object. Values of memory index equal to or below 0.5 indicate memory deficit. Data represent means ± SEM. Kruskal-Wallis test ** p* < 0.05, followed by Benjamini, Hochberg, and Yekutieli post-hoc: *** p* < 0.005, CS (white bar) v/s AS (red bar), ** p* < 0.05; AS v/s Asphyxia-DFX-MSC-S (cyan bar); ** p* < 0.05, AS v/s Asphyxia-TNFα+IFN-γ-MSC-S (green bar), *N* = 7–9 samples per group.

**Table 1 ijms-21-07800-t001:** Apgar scale evaluation. Data are expressed as means ± SEM. *n*, number of pups; *m*, number of dams. The effect of perinatal asphyxia (PA) was evaluated on (i) survival (*Mann Whitney test U* = 1220, *** *p* < 0.0001); (ii) body weight (g; n.s.); respiratory frequency (events/min) (*Mann Whitney test U* = 0.50, *p* < 0.0001); (iii) spontaneous movement (0–4; 0 = no movements; 4 = coordinated movements of fore and hindlimbs, as well as head and neck) (*Mann Whitney test U* = 0, *** *p* < 0.0001); (iv) vocalization % (yes/no) (*Mann Whitney test U* = 370, *p* < 0.0001); (v) gasping (yes/no) (*Mann Whitney test U* = 1000; *** p* < 0.005); color of skin (% pink, pink blue, or blue pink). * Comparisons with respect to controls (Bold); n.s., no statistically significant differences.

Parameters	Caesarean-Delivered (CS) Pups (*n* = 40, *m* = 13)	Asphyxia-Exposed (AS) Pups (*n* = 61, *m* = 13)
Survival (%)	100	**66.30** **± 5.0 *****
Body weight (g)	6.11 ± 0.07	5.98 ± 0.06
Respiratory frequency (events/min)	83.38 ± 1.4	**32.82 ± 1.6 *****
Spontaneous movements (0–4)	3.98 ± 0.03	**0.28 ± 0.07 *****
Vocalization (%)	97.5 ± 2.5	**27.87** **± 5.8 *****
Gasping (%)	0	**18.03** **± 4.96 ****
Skin color		
Pink (%)	100	**0**
Pink-Blue (%)	0	**83.61**
Blue-Pink (%)	0	**16.4**
Sex		
Female (%)	55.0	54.10
Male (%)	45.0	45.90

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
