# Peer review of "Intranasal Administration of Mesenchymal Stem Cell Secretome Reduces Hippocampal Oxidative Stress, Neuroinflammation and Cell Death, Improving the Behavioral Outcome Following Perinatal Asphyxia"

_ijms, 2020, doi:10.3390/ijms21207800_

Round 1

Reviewer 1 Report

The paper concerns an extremely interesting problem both from scientific and medical viewpoint. Therapeutic effects of intranasal MSC-S in asphyxiated newborn rats were evaluated with a wide variety of adequate measurements and assays.

In general the paper is well structured; the results are presented clearly and conclusions are justifiable.

Major concerns

In the methods sections the authors say that following asphyxia the pups were placed on a warming pad for approximately (why not exactly?!) 40 minutes but they (the authors) do not specify thermal conditions. Thermal conditions imposed on neonatal rats are of paramount importance for pathology of perinatal anoxia and of the subsequent reoxygenation stress. More precise is the description of the conditions during asphyxia i.e. the fetuses were immersed into a water bath at 37°C for 21 minutes. Their body temperature must have been then clamped at 38°C (the reference belows). Those were extremely severe thermal conditions indeed. No wonder that as many as 33.7% of pups died afterwards despite of their stimulation to respire (the method of the stimulation should also be shortly described).

I suggest at least short discussing the problem in the Discussion section, for reference see: Rogalska, J.; Caputa, M., Spontaneously reduced body temperature and gasping ability as a mechanism of extreme tolerance to asphyxia in neonatal rats. J. Thermal Biol. 2005, 30, 360-369.

Minor concerns

  1. Microglial reactivity (page 9, bottom paragraph), similar to other experimental variables, should be presented as a separate section.
  2. In Fig. 5 there is a lack of data concerning Asphyxia DFX MSC-S group and no comments on it.

Author Response

Manunscript, ID IJMS-934310,

Point-by-point response to the comments and criticisms raised by the Reviewers.

The reply explains the manner we have dealt with the requested revisions, according to your guide to writing a response to the reviewers' comments

Manuscript  IJMS-934310

September 30, 20202

ACTION/REPLY  TO REVIWER´S COMMENTS.

Thank you for your comments on the improvement of the manuscript.  Your earlier comments allowed us to make these modifications. 

Comments and Suggestions for Authors

Reviewer 1

C1: The paper concerns an extremely interesting problem both from scientific and medical viewpoint. Therapeutic effects of intranasal MSC-S in asphyxiated newborn rats were evaluated with a wide variety of adequate measurements and assays. In general, the paper is well structured; the results are presented clearly and conclusions are justifiable.

R1: We appreciate your kind comment, and we adhere below to your recommendations

C2: “In the methods sections the authors say that following asphyxia the pups were placed on a warming pad for approximately (why not exactly?!) 40 minutes but they (the authors) do not specify thermal conditions. Thermal conditions imposed on neonatal rats are of paramount importance for pathology of perinatal anoxia and of the subsequent reoxygenation stress. More precise is the description of the conditions during asphyxia i.e. the fetuses were immersed into a water bath at 37°C for 21 minutes. Their body temperature must have been then clamped at 38°C (the reference below). Those were extremely severe thermal conditions indeed. No wonder that as many as 33.7% of pups died afterwards despite of their stimulation to respire (the method of the stimulation should also be shortly described)”.

I suggest at least short discussing the problem in the Discussion section, for reference see: Rogalska, J.; Caputa, M., Spontaneously reduced body temperature and gasping ability as a mechanism of extreme tolerance to asphyxia in neonatal rats. J. Thermal Biol. 2005, 30, 360-369.”

 R2: Many thanks for your recommendation. We have now include a more precise description of perinatal asphyxia including the thermal conditions in the Material and Methods section (Section 4.3 Perinatal asphyxia, Line 843-854), and we also comment this issue in the Discussion section (Lines 631-634). The paper recommended was incorporated as a new reference.

We agree that thermal conditions imposed on neonatal rats are of paramount importance for pathology of perinatal anoxia and the subsequent reoxygenation stress. The perinatal asphyxia model used, mimics intrauterine thermal conditions occurring in a prolonged delivery. For that reason, the water temperature and the warming pad were set at 37 °C to obtain severe conditions found after neonatal asphyxia, required to show a clear treatment effect. While the water temperature had to be 37ºC, being the intrauterine environment; the 37ºC temperature of the warming pad, if lowered to 33 to 34ºC could have led to reduced injury, which would not render the marked neonatal dysfunction needed to test the therapeutic approaches studied. We trust that you will agree that the present study showed marked therapeutic effects in a model of clearly adverse pathological conditions

Materials and Methods. 4.3 Perinatal asphyxia

Pregnant Wistar rats in the last day of gestation (G22) were euthanized and hysterectomized. Three or four pups per dam were removed immediately to be used as non-asphyxiated cesarean-delivered controls. The remaining fetus-containing uterine horns were immersed in a water bath at 37 °C for 21 min, in order to maintain uterine thermal conditions [106]. Following asphyxia, uterine horns were incised and pups removed, stimulated to breathe by wiping the amniotic fluid and by tactile stimulation of the oral region with pieces of medical wipes as well as by pressing the thorax. This resuscitation maneuver implies expert and skillful handling, taking a long time (4–6 min) to stimulate the first gasp, and an even longer time to establish regular breathing, always supported by gasping. Then, after a 40 min observation period on a warming pad within the thermal neutral ranges 36-37 °C [106], the pups were evaluated with an Apgar scale adapted for rats, according to Dell’Anna et al. [107], randomly assigned to each experimental condition and nursed by a surrogate dam.

Discussion  Paragraph 2, Line Lines 631-634 “Temperature control after brain injury is critical to optimize recovery and functional outcome in neonatal rats (Rogalska & caputa 2005, Wood et al 2017).  Currently, therapeutic hypothermia is the only routinely used clinical intervention for full-term infants with PA. This intervention reduces death and disability, but it does not provide complete protection (Cambel et al., 2018)”.

C3: Microglial reactivity (page 9, bottom paragraph), similar to other experimental variables, should be presented as a separate section.

R3: Many thanks.  As recommended, microglial reactivity is now presented as a separated section (line 318-319). 2.1.5 Effect of intranasal administration of MSC-S on PA-induced microglial reactivity determined in hippocampus at P7.

C4: In Fig. 5 there is a lack of data concerning Asphyxia DFX MSC-S group and no comments on it.

R4We fully agree with the reviewer. Regrettably, the data on the effect of DFX-MSC on PA-induced microglial reactivity could not be fully gathered due to Government mandated restrictions in the country (and laboratories) in response to the global COVID-19 pandemic. However, all other parameters determined were essentially identical for both DFX-MSC-S and TNFa-IFNg-MSC-S, namely, (i) oxidative stress protection, (ii) anti-inflammatory effects, evaluated by a decrease in nuclear p65 levels, and (iii) antiapoptotic effects (decrease in cleaved caspase-3). Thus, it is expected that microglial reactivity PA-induced would also be reversed by DFX-MSC-S treatment. Given the present experimental conditions, we trust that this omission might be considered only a relative design weakness.

This information is included in in the discussion section  ( Paragraph 3, lines 635-647;771-786)

Discussion: Line 635-647. The present study shows that intranasal administration to asphyxia exposed rat pups of secretome derived from MSC preconditioned with either DFX or TNF-a+IFN-g:  (i) fully suppressed the long-lasting hippocampal oxidative stress (elevated GSSG/GSH ratio) observed following PA at P7 and P60; (ii) induced NRF2 nuclear translocation and decreased NRF2 levels in cytoplasm at P7; (iii) increased the NQO1 antioxidant protein levels at P7; (iv) decreased p65 associated neuroinflammation and microglial reactivity at P7; (v) decreased cleaved-caspase-3 protein levels (a critical executioner of apoptosis) at P7; (v) improved behavioral development evaluated by the righting, negative geotaxis and cliff aversion reflexes, motor coordination, locomotion, decreased anxiety, improved recognition memory deficits; (vi) MSC-S preconditioned with either DFX or TNF-α+IFN-γ, showed similar effectiveness to prevent the PA-induced deleterious effects. However, DFX- but not TNF-α+IFN-g-preconditioned MSC-S increased NQO1 protein levels.

Lines 771-786

The antagonism between NRF2 and p65 can occur in the CNS, where astrocytes are the main protagonists. In the absence of NRF2, astrocytes induce inflammation via p65 activation, leading to downstream pro-inflammatory cytokine transcription [100]. This could also imply microglia since their morphological changes decreasing the length of primary and secondary processes in stratum radiatum of CA1 hippocampus of asphyxia-exposed rats reflect an inflammatory phenotype. This phenotype was reverted by TNF-α+IFN-γ-MSC-S, as shown by this study. In the neonatal HI model at P10, human umbilical cord tissue-derived MSC administered intranasally induced beneficial effects observed in hippocampus, namely a decrease of neuronal loss and a reduction of neuroinflammation markers, including a decrease of activated microglia and astrocyte density [101]. Furthermore, in a transgenic Alzheimer´s disease mice model, the treatment with exosomes from hypoxic-preconditioned MSC decreased the activation of astrocytes and microglia, promoted the transformation of microglia to dendritic cells and down-regulated pro-inflammatory cytokines (TNF-a and IL-1b) in mice hippocampus [102], suggesting that the preconditioning of MSC with other hypoxia mimetic molecules, such as DFX could also prevent glial reactivity, an issue that remains to be further investigated.

Reviewer 2 Report

Title: Intranasal administration of preconditioned mesenchymal stem cell secretome cell death in hippocampus and improved behavioral performance following perinatal asphyxia

Comments:

  1. Title: the title is too long and it is not a good sentence grammatically. Please revise it. You may consider something like “Intranasal administration of preconditioned mesenchymal stem cell secretome reduces hippocampal cell death  and improved behavioral performance following perinatal asphyxia”
  2. Results: The Apgar evaluation should also be compared in the 3 asphyxia-exposed groups: asphyxia-Vehicle, asphyxia-DFX-MSC-s and asphyxia -TNF-+IFN-MSC-s.
  3. Did all the asphyxia animals survive?
  4. Figure 5: only one treatment group was included (asphyxia -TNF-+IFN-MSC-s)
  5. In the Table, and all the figures (figure legends and text): with a small sample size (most of the time less than 10), p < 0.005, p <0.0005 and p < 0.0001 are not statistically meaningful. The significance level should be kept at p < 0.05.
  6. Discussion: the term “hypoxia-induced protection” is confusing, since the hypoxia can induce damages such as the impairment demonstrated in the asphyxia model. Hypoxic preconditioning is protective. I think using hypoxic preconditioning is more appropriate.
  7. Conclusion: “In conclusion, intranasal administration of MSC-S to PA-exposed rats decreased oxidative stress, neuroinflammation and cell death, improving neurobehavioral development, motor coordination, locomotor activity, cognition, and decreasing anxiety. Secretome derived from MSC, preconditioned with either DFX or TNF-+IFN- yielded similar effects, improving the PA outcome.” However, in the results, only the MSC-S preconditioned with either DFX or TNF-+IFN were demonstrated, there was no MSC-S (without preconditioning) group. Then it is hard to tell if the protection is from MSC-s or the preconditioning (DFX or TNF-+IFN) or both.
  8. There are too many grammatical and syntax errors, professional editing and proof reading will be helpful.
  9. Minor: Abstract: “reversed hippocampal asphyxia-induced oxidative stress”, it should be “reversed asphyxia-induced oxidative stress in hippocampus”. Abstract: “NRF2” should be spelled out at first use.

Author Response

Manuscript  IJMS-934310

September 30, 20202

 ACTION/REPLY  TO REVIWER´S COMMENTS.

Thank you for your comments on the improvement of the manuscript.  Your earlier comments allowed us to make these modifications. 

 Comments and Suggestions for Authors

Reviewer 2

C1: Title: Intranasal administration of preconditioned mesenchymal stem cell secretome cell death in hippocampus and improved behavioral performance following perinatal asphyxia

Title: the title is too long and it is not a good sentence grammatically. Please revise it. You may consider something like “Intranasal administration of preconditioned mesenchymal stem cell secretome reduces hippocampal cell and improved behavioral performance following perinatal asphyxia”

R 1: Many thanks for your suggestion. We have shortened the title as follow:

 Intranasal administration of mesenchymal stem cell secretome reduces hippocampal oxidative stress, neuroinflammation and cell death, and improves behavioral deficits following perinatal asphyxia” 

death

C2: Results: The Apgar evaluation should also be compared in the 3 asphyxia-exposed groups: asphyxia-Vehicle, asphyxia-DFX-MSC-S and asphyxia -TNF-+IFN-MSC-S.   Did all the asphyxia animals survive? 

R2Many thanks. We now explain that the Apgar evaluation was performed 40 minutes after birth on controls and asphyxia exposed rats. Thereafter, controls and asphyxia-exposed rats were randomly assigned to each experimental condition (line 175; 182-184). The values of respiratory frequency (event/min) for asphyxia-Vehicle, asphyxia-DFX-MSC-S and asphyxia -TNF-+IFN-MSC-S groups were 30.53 ± 2.39, 29.26 ± 2.46 and 30.15 ± 3.91event/min, respectively, showing a homogeneous distribution in experimental groups of the pups.

The rate of survival shown by PA neonates was approximately 66%, while it was 100% among controls. Surviving asphyxia-exposed rats showed decreased respiratory frequency (~60%), decreased vocalization (~70%), blue (cyanotic) skin, rigidity and akinesia, compared to sibling control rats, indicating a severe insult (lines 178-182). Animal survival is also indicated (Table 1 line 192). After intranasal administration of vehicle or preconditioned MSC-S all the animals survived (line 182-184). 

C3: Figure 5: only one treatment group was included (asphyxia -TNF-+IFN-MSC-s)

R3R4We fully agree with the reviewer. Regrettably, the data on the effect of DFX-MSC on PA-induced microglial reactivity could not be fully gathered due to Government mandated restrictions in the country (and laboratories) in response to the global COVID-19 pandemic. However, all other parameters determined were essentially identical for both DFX-MSC-S and TNFa-IFNg-MSC-S, namely, (i) oxidative stress protection, (ii) anti-inflammatory effects, evaluated by a decrease in nuclear p65 levels, and (iii) antiapoptotic effects (decrease in cleaved caspase-3). Thus, it is expected that microglial reactivity PA-induced would also be reversed by DFX-MSC-S treatment. Given the present experimental conditions, we trust that this omission might be considered only a relative design weakness.

This information is included in in the discussion section,  Paragraph 3, lines 635-647;771-786.

Discussion: Line 635-647. The present study shows that intranasal administration to asphyxia exposed rat pups of secretome derived from MSC preconditioned with either DFX or TNF-a+IFN-g:  (i) fully suppressed the long-lasting hippocampal oxidative stress (elevated GSSG/GSH ratio) observed following PA at P7 and P60; (ii) induced NRF2 nuclear translocation and decreased NRF2 levels in cytoplasm at P7; (iii) increased the NQO1 antioxidant protein levels at P7; (iv) decreased p65 associated neuroinflammation and microglial reactivity at P7; (v) decreased cleaved-caspase-3 protein levels (a critical executioner of apoptosis) at P7; (v) improved behavioral development evaluated by the righting, negative geotaxis and cliff aversion reflexes, motor coordination, locomotion, decreased anxiety, improved recognition memory deficits; (vi) MSC-S preconditioned with either DFX or TNF-α+IFN-γ, showed similar effectiveness to prevent the PA-induced deleterious effects. However, DFX- but not TNF-α+IFN-g-preconditioned MSC-S increased NQO1 protein levels.

Lines 771-786: The antagonism between NRF2 and p65 can occur in the CNS, where astrocytes are the main protagonists. In the absence of NRF2, astrocytes induce inflammation via p65 activation, leading to downstream pro-inflammatory cytokine transcription [100]. This could also imply microglia since their morphological changes decreasing the length of primary and secondary processes in stratum radiatum of CA1 hippocampus of asphyxia-exposed rats reflect an inflammatory phenotype. This phenotype was reverted by TNF-α+IFN-γ-MSC-S, as shown by this study. In the neonatal HI model at P10, human umbilical cord tissue-derived MSC administered intranasally induced beneficial effects observed in hippocampus, namely a decrease of neuronal loss and a reduction of neuroinflammation markers, including a decrease of activated microglia and astrocyte density [101]. Furthermore, in a transgenic Alzheimer´s disease mice model, the treatment with exosomes from hypoxic-preconditioned MSC decreased the activation of astrocytes and microglia, promoted the transformation of microglia to dendritic cells and down-regulated pro-inflammatory cytokines (TNF-a and IL-1b) in mice hippocampus [102], suggesting that the preconditioning of MSC with other hypoxia mimetic molecules, such as DFX could also prevent glial reactivity, an issue that remains to be further investigated.

C4: In the Table, and all the figures (figure legends and text): with a small sample size (most of the time less than 10), p < 0.005, p <0.0005 and p < 0.0001 are not statistically meaningful. The significance level should be kept at p < 0.05. 

R4: The significance level was indicated according to the results obtained in the applied statistical test (Prism Software). Indeed, while significance does not change, the probability of repeating the differences reported is inferred by the actual p value. We trust that you will accept our description.

C5: Discussion: the term “hypoxia-induced protection” is confusing, since the hypoxia can induce damages such as the impairment demonstrated in the asphyxia model. Hypoxic preconditioning is protective. I think using hypoxic preconditioning is more appropriate.

R5: Many thanks for your suggestion; the phrase was changed as recommended ( In Discussion, paragraph 4, line 659-660).

C6: Conclusion: “In conclusion, intranasal administration of MSC-S to PA-exposed rats decreased oxidative stress, neuroinflammation and cell death, improving neurobehavioral development, motor coordination, locomotor activity, cognition, and decreasing anxiety. Secretome derived from MSC, preconditioned with either DFX or TNF-+IFN- yielded similar effects, improving the PA outcome.” However, in the results, only the MSC-S preconditioned with either DFX or TNF-+IFN were demonstrated, there was no MSC-S (without preconditioning) group. Then it is hard to tell if the protection is from MSC-s or the preconditioning (DFX or TNF-+IFN) or both.

R6: Many thanks.  We agree and have made sure that this is not misinterpreted (Conclusion: Line 1021-1026)

In conclusion, intranasal administration to PA-exposed rats of the secretome derived from MSC, preconditioned with either DFX or TNF-a-+IFN-g, decreased oxidative stress, neuroinflammation and, cell death, improving neurobehavioral development, motor coordination, locomotor activity, cognition, and decreasing anxiety. The secretome derived from MSC, preconditioned with either DFX or TNF-a+IFN-g, yielded similar effects, improving the PA outcome.

C7: There are too many grammatical and syntax errors, professional editing and proof reading will be helpful.

R7: Our apologies for the grammatical errors, the revised manuscript has been thoroughly checked and trust it has reduced any shortcomings (labeled text).

C8: Abstract: “reversed hippocampal asphyxia-induced oxidative stress”, it should be “reversed asphyxia-induced oxidative stress in hippocampus”. Abstract: “NRF2” should be spelled out at first use.

R8: Many thanks. The sentences have been improved as recommended (line 37-39).

Reviewer 3 Report

This manuscript by Farfan and colleagues, is an extensive analyses of the effects of preconditioned MSC-S following perinatal asphyxia. The data is expansive and the manuscript is overall well-written. The data is well done, however lack of primary methodological detail precludes complete assessment of scientific soundness. The length of the introduction and discussion precludes complete clarity and detracts from the massive amounts of data at hand. The authors would be better served by streamlining their intro and discussion to read less like a review article. Despite these critiques and the those listed below, the manuscript addresses an important topic from a scientific and clinical standpoint and highlights the importance of the stem cell secretome as a beneficial milieu for neurorepair.

Major Points:

  1. The title suggests the repair of cognition, although object recognition memory and exploratory behavior was what was assayed. Might it not be more accurate to name these behaviors specifically as cognition encompasses many domains? The use of a memory index is confusing and it is unclear why raw data is not presented. Inclusion of raw data as opposed to a memory score would strengthen significantly.
  2. The authors rely too heavily on prior citations and more methodological detail needs to be provided to ensure reproducibility and enhance rigor. For example, it is stated that unbiased stereology was used but no details are provided. How were the brains sectioned? At what intervals? How thick? What were the anatomical boundaries of the reference space? What were the z-stacks used and how was the immunohistochemistry performed? The protocol of the neurite tracing should be described at least briefly. With the details provided it is impossible to determine whether the technique used was either unbiased or stereology.
  3. It is unclear how the secretome can be 6 micrograms. It would be helpful to describe how this was calculated or assayed, as the secretome is typically liquid. Was protein assayed to come up with this value? Similarly, how was it determined that 16ul was sufficient volume? How were the number of cells used to generate the secretome determined?
  4. Is perinatal asphyxia the same as profound hypoxia-ischemia or hypoxic-ischemic encephalopathy (HIE)? What clinical scenario are the authors recapitulating? Assuming the authors are replicating a term infant brain injury scenario it would aid the readers to provide these details. How does PA occur? Uterine rupture? Cord Prolapse? As rats are third trimester equivalent at birth how should the reader frame this model in a clinical scenario? These details should be provided.

Minor Points:

  • The authors may consider referring to their test subjects as neonatal rats or rats or rat pups to avoid confusion with human neonates. Similarly it should be stated that a rat APGAR scale or APGAR scoring scale adapted for rats were used.
  • As the methods are presented at the end of the manuscript it would aid the reader if the ages of rats were described upon first presentation (i.e. what age neonatal rats) and if the type of MSCs used in the cultures were stated up front (human derived adipose MSCs).

Author Response

Manunscript, ID IJMS-934310,

Point-by-point response to the comments and criticisms raised by the Reviewers.

The reply explains the manner we have dealt with the requested revisions, according to your guide to writing a response to the reviewers' comments

September 30, 20202

ACTION/REPLY  TO REVIWER´S COMMENTS.

Thank you for your comments on the improvement of the manuscript.  Your earlier comments allowed us to make these modifications

Reviewer 3

C1: This manuscript by Farfan and colleagues, is an extensive analyses of the effects of preconditioned MSC-S following perinatal asphyxia. The data is expansive and the manuscript is overall well-written. The data is well done, however lack of primary methodological detail precludes complete assessment of scientific soundness. The length of the introduction and discussion precludes complete clarity and detracts from the massive amounts of data at hand. The authors would be better served by streamlining their intro and discussion to read less like a review article. Despite these critiques and the those listed below, the manuscript addresses an important topic from a scientific and clinical standpoint and highlights the importance of the stem cell secretome as a beneficial milieu for neurorepair.

 R1: We appreciate your kind assessment.

C2: The title suggests the repair of cognition, although object recognition memory and exploratory behavior was what was assayed. Might it not be more accurate to name these behaviors specifically as cognition encompasses many domains? The use of a memory index is confusing and it is unclear why raw data is not presented. Inclusion of raw data as opposed to a memory score would strengthen significantly.

R2:  Many thank, we agree with your comment. Memory deficit instead of the cognitive deficit is more appropriate. The title was changed to: “Intranasal administration of mesenchymal stem cell secretome reduces hippocampal oxidative stress, neuroinflammation and cell death and improves behavioral deficits following perinatal asphyxia” 

The memory recognition index represents the proportion of time by which the animal explores the new object compared to the old one, as a proportion of the total exploration time. This index is the original (a) and widely used way to measure recognition memory (b). The use of raw measures such as exploration time is discouraged; every animal can display a different total amount of exploration time, but a similar proportion of new/old object inspection.

a)  Sivakumaran et al., 2018 (https://www.nature.com/articles/s41598-018-30030-7)

b) Ennaceur, A. & Delacour, J. A new one - trial test for neurobiological studies of memory in rats. 1″ Behavioral data. Behav. Brain Res. 31, 47–59 (1988)

C3: The authors rely too heavily on prior citations and more methodological detail needs to be provided to ensure reproducibility and enhance rigor. For example, it is stated that unbiased stereology was used but no details are provided. How were the brains sectioned? At what intervals? How thick? What were the anatomical boundaries of the reference space? What were the z-stacks used and how was the immunohistochemistry performed? The protocol of the neurite tracing should be described at least briefly. With the details provided it is impossible to determine whether the technique used was either unbiased or stereology.

R3: Many thanks. We have greatly extended this section in Methods. Please see below and in the labelled text (4.11. Microglia reactivity lines 949 to 957).

Briefly, controls and asphyxia-exposed rats at P7 were anesthetised and perfused intracardially with 0.1M PBS (pH 7.4), followed by a formalin solution (4% paraformaldehyde in 0.1M PBS, pH 7.4). The brain was removed from the skull, post-fixed in the same fixative solution overnight at 4°C and immersed in 10% sucrose containing 0.1 M PBS for 48h and subsequently in 30% sucrose at 4°C for 48hs. Coronal sections (30 μm thick) of the hippocampi (between Bregma -1.40 and -2.00 mm, https://www.ial-developmental-neurobiology.com/en/publications/collection-of-atlases-of-the-rat-brain-in-stereotaxic-coordinates) were obtained using a cryostat (Thermo Scientific Microm HM 525, Germany). Sections were rinsed with 0.1M PBS and treated with blocking solutions for 1 h, incubated with primary antibody against IBA-1 (rabbit, Wako Chemicals USA, #019-19741; 1:500 in blocking solution containing 1% BSA, 10% NGS and 0.3% Triton X 100) overnight at 4°C. After repeated rinsing with 0.1M PBS, sections were incubated with the secondary antibody anti-rabbit-Alexa 488 (goat, Thermo Fisher Scientific, IL, USA, #A-11034; 1:500 incubated in blocking solution containing 1% NGS) and counterstained with DAPI (Thermo Fisher Scientific, # D1306, 0.02 M; 0.0125 mg/ml; for nuclear labelling) for 2 h. After rinsed, the samples were mounted on silanized slides with Fluoromount-GTM (Thermo Fisher Scientific) and examined by confocal microscopy (Olympus-fv10i, Center Valley, PA, USA).  

Microphotographs (five–six) were taken from stratum radiatum of CA1 hippocampal region in the field of an Olympus FV10i confocal microscope (Center Valley, PA, USA), using 60× objective lens (NA1.30). Images were captured using FV10-ASW-2b software (Olympus). The area inspected for each stack was 0.04 mm2 and the thickness (Z axis) was measured for each case. The total length of primary and secondary processes for six IBA-1 positive cells per Z-stack, were analyzed by three-dimensional reconstruction using the Simple Neurite Tracer plugin of FIJI image analysis (http://fiji.sc/Simple_Neurite_Tracer)  [99] following the protocol described by Tavares et al [100]

C4: It is unclear how the secretome can be 6 micrograms. It would be helpful to describe how this was calculated or assayed, as the secretome is typically liquid. Was protein assayed to come up with this value? Similarly, how was it determined that 16ul was sufficient volume? How were the number of cells used to generate the secretome determined?

R4: Each secretome dose was obtained from 2x105 preconditioned MSC. For secretome production, the culture medium obtained from preconditioned cells was centrifuged at 400 x g for 10 minutes to remove whole cells and passed through a 0,22 mm filter to remove protein aggregates. The filtrate was concentrated 50 times using 3 kDa cutoff filters (Millipore, Midlesex, MA). Protein concentration in the final secretome was determined using the BCA protein assay kit (Thermo Scientific, Waltham, MA).  The 16 ml volume used in this study was determined by scaling-down (1/10 v/v) the previously reported dose used for intranasal administration of secretome in adult rats (Quintanilla et al, 2019). We have now added this information to the Methods section (4.5. Preconditioning of human adipose tissue-derived MSC and secretome generation,lines 873 to 881)

C5: Is perinatal asphyxia the same as profound hypoxia-ischemia or hypoxic-ischemic encephalopathy (HIE)? What clinical scenario are the authors recapitulating? Assuming the authors are replicating a term infant brain injury scenario it would aid the readers to provide these details. How does PA occur? Uterine rupture? Cord Prolapse? As rats are third trimester equivalent at birth how should the reader frame this model in a clinical scenario? These details should be provided.

R5: Many thank you for the recommendation. Now we includes these aspects in the introduction section (line 51-56, 60-62) and in the discussion section (Lines 618 to 630) .Actuality, there is not a consensus on a reliable and predictable experimental model for the study of perinatal asphyxia. While the present model is clearly a hypoxia-reoxygenation model other common  experimental models investigate the issue at postnatal stages, as proposed by Vannucci in 1981, corresponding to perinatal hypoxia-ischemia model induced by common carotid artery ligation ( ischemia) and hypoxemia at P7. While Perinatal Asphyxia model, proposed at the Karolinska Institutet, Stockholm, Sweden, by Bjelke et al. (1991), corresponding to perinatal hypoxia induced at delivery time reflecting a hypoxia-reoxygenation model. Both represent different clinical scenarios. In the model utilized in this study, asphyxia occurs by alteration in gas exchange, followed by hypoxemia, acidosis and hypercapnia, mandatory criteria for a clinically relevant model of perinatal asphyxia (Herrera-Marschitz 2014). Furthermore, the model used in this study is not accompanied by ischemia, therefore the brain damage is global, and other organs with high oxygen demand are also affected (Herrera-Marschitz et al.2011). 

 Although the temporality has been criticized, arguing that the brain of neonate rats is premature when compared to the neonatal human brain, a statement mainly referring to the neocortex (Romijn et al., 1991),  our view is that the degree of maturity depends upon the tissue and function selected for the comparisons, the vulnerability is relating to both the timing and the location of the insult (Craig et al., 2003).

Description of Global Perinatal Asphyxia models: At the time of delivery, a first spontaneous delivery can be observed before the dams are neck dislocated and subjected to a caesarean section and hysterectomy. The uterine horns containing the fetus are immediately immersed into a water bath at 37°C for 21 min. Following asphyxia, the pups are removed from the uterine horns and resuscitated. Additional efforts and care are taken to induce and maintain pulmonary breathing. (see Herrera-Marschitz et al. 2014).

Herrera-Marschitz, et al . Perinatal asphyxia: current status and approaches towards neuroprotective strategies, with focus on sentinel proteins. Neurotox. Res. 19(4), 603-627 (2011).

Herrera-Marschitz, et al. Perinatal asphyxia: CNS development and deficits with delayed onset. Frontiers in Neuroscience. 8,47 (2014). 

Vannucci, R.G., Vannucci, S.J. Perinatal hypoxic-ischemic brain damage evolution of an animal model. Dev. Neurosci. 27, 81-86 (2005).

Bjelke, B. et al Asphyctic lesion: proliferation of tyrosine hydroxylase immunoreactivity nerve cell bodies in the rat substantia nigra and functional changes in dopamine neurotransmission. Brain Res. 543,1–9.(1991)

 Romijn et al). At what age is the developing cerebral cortex of the rat comparable to that of the full-term newborn human baby? Early Hum. Dev. 26, 61–67 10.1016/0378-3782(91)90044-4 (1991

Craig A., et al. Quantitative analysis of perinatal rodent oligodendrocyte lineage progression and its correlation with human. Exp. Neurol. 181, 231–240 10.1016/S0014-4886(03)00032-3. (2003). 

C6:  Minor points. The authors may consider referring to their test subjects as neonatal rats or rats or rat pups to avoid confusion with human neonates. Similarly, it should be stated that a rat APGAR scale or APGAR scoring scale adapted for rats were used.

R6: Many Thanks.  As recommended, this information was incorporated. Please see in the marked-up text (line 174, 175, 178, 184….)

C7:  As the methods are presented at the end of the manuscript it would aid the reader if the ages of rats were described upon first presentation (i.e. what age neonatal rats) and if the type of MSCs used in the cultures were stated up front (human derived adipose MSCs).

R7:  Many thanks. This information is now incorporated (lines 28, 175, 194, 214,230….).

Round 2

Reviewer 2 Report

The authors answered the most of reviewers' questions, 

Minor: 

Line 58: “mayor”? :

Conclusion “In conclusion, intranasal administration to PA-exposed rats of the secretome derived from MSC, preconditioned with either DFX or TNF-a-+IFN-g, decreased oxidative stress, neuroinflammation and, cell death, improving neurobehavioral development, motor coordination, locomotor activity, cognition, and decreasing anxiety. “ This sentence needs to be rephrased.

Author Response

Thank you for your comments, improving the quality of the manuscript.

Reviewer 2

C1: The authors answered the most of reviewers' questions, 

R1: Thanks for your comments helping us to improve the manuscript. We are adhering to your recommendations.

C2: Minor: Line 58: “mayor”?

R2: Our apologies for the grammatical error. “Mayor” was changed by major in the revised manuscript (Line 56).

C3: Conclusion “In conclusion, intranasal administration to PA-exposed rats of the secretome derived from MSC, preconditioned with either DFX or TNF-α-+IFN-γ, decreased oxidative stress, neuroinflammation, and cell death, improving neurobehavioral development, motor coordination, locomotor activity, cognition, and decreasing anxiety. “This sentence needs to be rephrased.

R3: Many thanks for your suggestion; we rephrased the conclusion as follow:

“In conclusion, intranasal administration of preconditioned MSC-derived secretome to PA-exposed rats decreased oxidative stress, neuroinflammation and cell death, improving the neurobehavioral development, motor coordination, locomotor activity, cognition, and decreasing anxiety. The secretome yielded similar effects whether the MSC were preconditioned with either DFX or TNF-α-+IFN-γ, improving the PA outcome. Thus, intranasal administration of preconditioned MSC-S is a novel therapeutic strategy to prevent the short- and long-term effects of perinatal asphyxia”. 

Reviewer 3 Report

The authors are complimented on their thorough and comprehensive revision.

Minor point:

Line 55 should be "psychiatric" and not "psychotic". 

Author Response

Thank you for your comments, improving the quality of the manuscript.

Reviewer 3

C1: The authors are complimented on their thorough and comprehensive revision.

R1: We appreciate your kind comment, and we adhere below to your recommendations.

C2: Minor point: Line 55 should be "psychiatric" and not "psychotic". 

R2: Many thanks for your suggestion; “psychotic” was changed as recommended (Line 53).